

# Modeling the effect of free convection on permafrost melting-rates in frozen rock-clefts

Amir Sedaghatkish[1,2], Frédéric Doumenc[3,4], Pierre-Yves Jeannin[1], Marc Luetscher[1]

[1]Swiss Institute for Speleology and Karst Studies (SISKA), CH-2300, La Chaux-de-Fonds, Switzerland
[2]Center for Hydrogeology and Geothermics (CHYN), University of Neuchâtel, 2000 Neuchâtel, Switzerland
[3]Université Paris-Saclay, CNRS, FAST, 91405, Orsay, Rue André Rivière, France
[4]Sorbonne Université, UFR 919, 4 place Jussieu, F-75252, Paris Cedex 05, France

*Correspondence to*: Amir Sedaghatkish (amir.sedaghatkish@isska.ch)

**Abstract**. Fully coupled heat transfer modeling during the thawing of frozen rock clefts yields melting rates that differ from those predicted by conventional conduction-based models. This research develops a conceptual model of a karst system subject to mountain permafrost supported by a numerical simulation incorporating free water convection. The numerical simulations rely on the apparent heat capacity method and the Darcy approach for energy and momentum equations. Notably, the anomalous behavior of water between 0 and 4 ℃ causes warmer meltwater to flow downwards, increasing the melting rate by approximately an order of magnitude as compared to conventional models that disregard free convection. The model outcomes are compared qualitatively with field data from Monlesi ice cave (Switzerland) and confirm the close agreement between the proposed model and real-world observations.

**Keywords**: Numerical modeling, free convection, melting rate, conduction, permafrost, karst, cave, hydrogeology

## 1 Introduction

With global climate change and rising temperatures, permafrost degradation has become a significant concern (Duvillard et al., 2021). This is particularly true in mountain regions where rapid thawing causes subsidence and rockfalls impacting construction works and tourist facilities (Haeberli et al., 2017). Thawing permafrost poses serious challenges to infrastructures built on frozen ground, including buildings, roads, pipelines, and railways. (Larsen et al., 2008; Cheng, 2005; Pham et al., 2008; Zhang et al., 2005; Fortier et al., 2011). Permafrost acts as a natural barrier, preventing water from infiltrating into the ground. As it thaws, drainage patterns are altered, leading to increased erosion and discharge of groundwater to rivers and lakes (Bense et al., 2012; Andresen et al., 2020; Fabre et al., 2017; Painter et al., 2016; Walvoord and Kurylyk, 2016). Permafrost degradation can also disrupt ecosystems adapted to frozen conditions. Trees, plants, and wildlife that rely on





permafrost stability may struggle to adapt to climate changes, leading to shifts in species distribution and ecosystem dynamics depending on the permafrost melting rate (Hayward et al., 2018; Krumhansl et al., 2015; Pelletier et al., 2018).

Field measurements in boreholes (e.g. Noetzli et al., 2021; Haberkorn et al., 2021) and caves (Luetscher et al., 2008; Colucci and Guglielmin, 2019; Wind et al., 2022) have shown that heat advected by water and air fluxes may significantly disturb the geothermal field, challenging classical models of heat propagation based on conductive fluxes. In carbonate environments, in

particular, the infiltration of water along well-developed conduits may develop a thermal anomaly deep into the karst system. The discontinuous nature of this permafrost may lead to the formation of massive cave ice at depth (Bartolomé et al., 2022) but also explain unexpected speleothem formations during glacial times (Luetscher et al., 2015; Fohlmeister et al., 2023; Fohlmeister et al., 2023)

Efforts are being made to study and understand permafrost degradation to mitigate its impacts. So far, most studies considered

mainly conductive and latent heat fluxes (Malakhova, 2022; Galushkin, 1997; Ivanov and Rozhin, 2022; Marchenko et al., 2008; Schuster et al., 2018; Jafarov et al., 2012; Cicoira et al., 2019; Hornum et al., 2020). Such models are, however, not applicable to heterogeneous, ice-rich media including debris-slopes, fractured and/or karst aquifers. But, even though conventional 1D transient models are not suitable to every context, they offer several advantages including lower computational costs and easy implementation. Pruessner et al., (2021) investigated glacier runoff associated with permafrost

degradation in high Alpine catchments. They used two different methods GERM (Huss et al., 2008) and SNOWPACK (Bartelt and Lehning, 2002) which are based on 1D transient conduction considering latent heat exchanges, different thermal properties of ground layers constituents and ventilation effects. This distributed model is efficient for large domains (catchment scale). Mohammed et al., (2021) developed a hydro-thermal-solutal model for analyzing reactive solute transport in permafrost-affected groundwater system by considering convective flux in the energy balance. Tubini et al., (2021) proposed a numerical

approach for modeling heat transfer of permafrost thawing in 1D domain which is capable to deal with high time steps and maintains conservation of energy in long-term simulations. Hasler et al., (2011) investigated the effect of advective heat transport in frozen rock clefts and fractures at small scales. They built a conceptual model combining numerical modeling and laboratory experiments. The effect of water flow inside the clefts is noticeable because of creating thermal shortcut between atmosphere and subsurface.

Although, the majority of these studies are designed to address large-scale problems they typically neglect free convection within the water phase. The authors nonetheless believe this process can play an important role in the total heat transfer particularly in geographically restricted areas subject to mountain permafrost (Haberkorn et al., 2021).

Our aim is to study the effect of free convection of water on the melting rate in frozen rock clefts and/or karst conduits at daily scale. Because of the increasing density of water between 0 and 4°C, density driven flow may be triggered by atmospheric

warming at the upper boundary. To assess the thermal reaction time, and thus hydrological breakthrough associated with the thawing of mountain permafrost we model the heat exchange in a frozen rock cleft. We prioritize the determination of the temperature field in the meltwater, ice, and surrounding rock, as well as the velocity field within the water domain to elucidate the mechanisms underlying the thawing of ice. Although a physical monitoring of such processes is hardly possible, our model



fits as close as possible to effective field observations. A systematic comparison is conducted under similar thermal settings
for two cases: one considers a conventional model including only conductive fluxes whilst the second case allows for free
water convection. Our study shows how free convection enhances the melting rate of an ice cleft at difference aperture sizes.

## 2 Computational domain and governing equations

Numerical modeling of free convection during ice melting is well-known in the field of computational fluid dynamics. The
density variations of water results in convective heat fluxes which may impact the melting rate of ice and, thus, control the
permafrost active layer. To model free convection, the momentum equations should be considered together with the energy
balance. There are numerous numerical approaches modeling latent heat and the velocity transitions between solid (ice) and
liquid (water) phases which can be classified as: 1) source term methods, 2) enthalpy methods and, 3) temperature-transforming
models (Nazzi Ehms et al., 2019). Similarly, to illustrate the transition in velocity between the solid and liquid phase in
momentum equations, three main groups of methods exist: 1) switch-off methods, 2) source term methods and, 3) variable
viscosity methods and there are further subcategories which can differ in their details (Nazzi Ehms et al., 2019; Caggiano et
al., 2018). In principle, any combination of two methods is applicable but, in this study, temperature-transforming models as
well as source term methods are selected for modeling velocity transition and latent heat, respectively (Neda and Nazif, 2020).
Figure 1-a illustrates the outlet of a frozen cleft hosted in an Alpine karst. Such clefts and fissures are characterized by distinct
geometries accommodating contrasted volumes of ice. To model the free convection of water resulting from the melting of
ice, a 2D computational domain comprising an ice-filled cavity surrounded by impermeable solid rock is selected (Fig. 1-b).
The conceptual model is represented in Fig. 1-b.

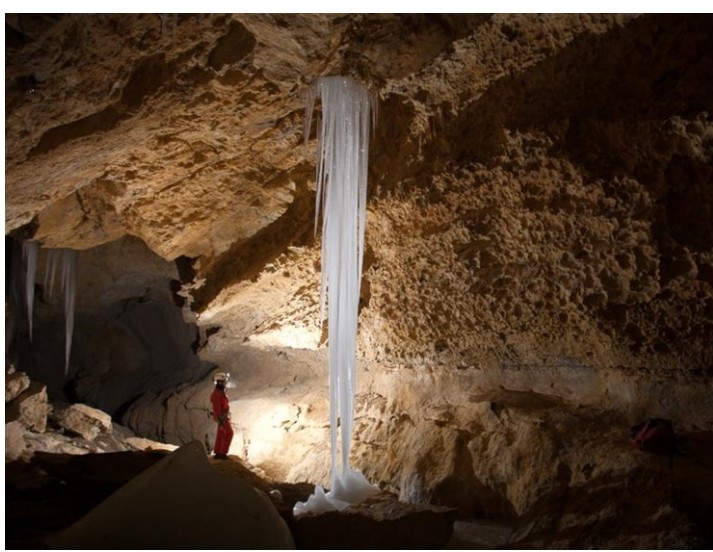



a)

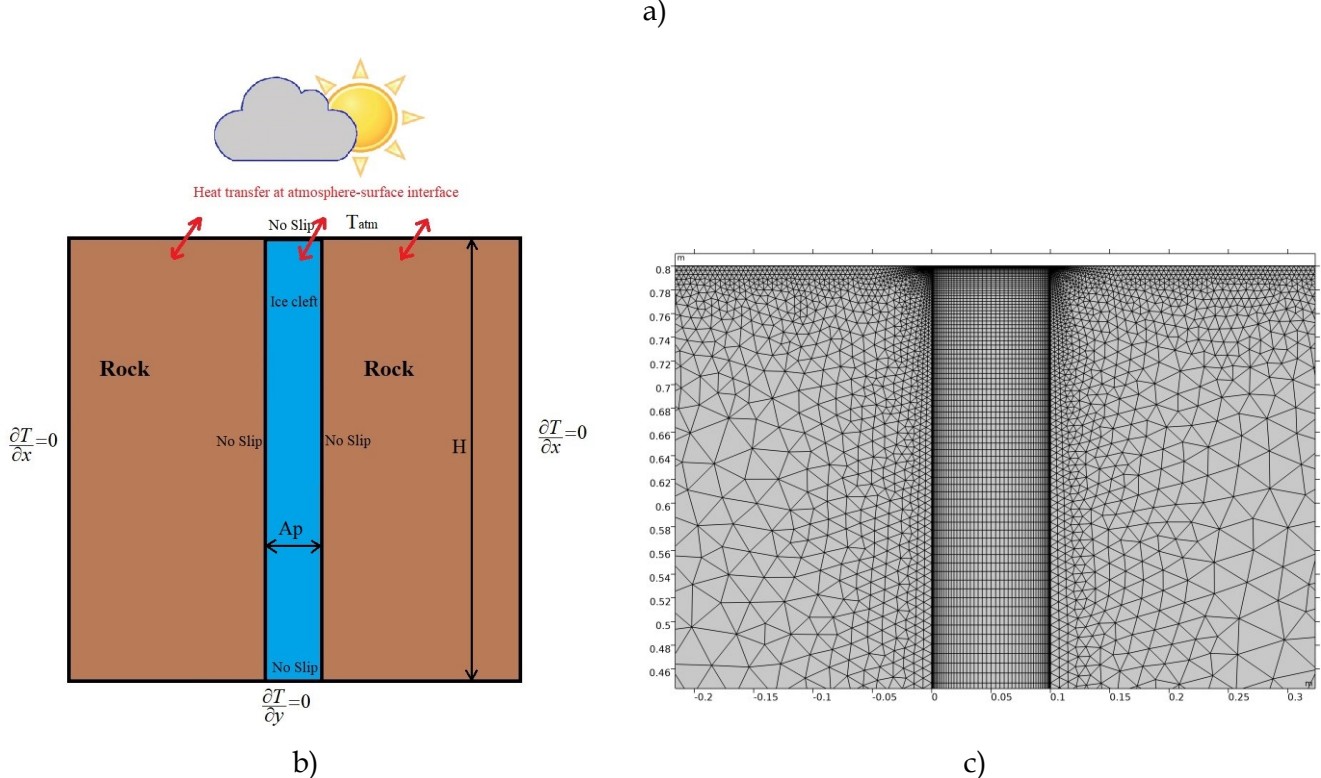

b)

c)

**Figure 1. a) Pingouins cave, Switzerland, showing ice-filled clefts exposed on the cave roof (Photo courtesy A. Conne) b) Computational domain and boundary conditions H=80 cm, A$_P$=10 cm c) Computational mesh close to upper part of ice-clef and surrounding rock**

The atmosphere temperature increases from -1 to 15 °C in 9 hours (equivalent to 1.77 °C/hr temperature increment rate) and it

melts the ice from top surface. This temperature increase is similar to the daily warming between the early morning and the afternoon. Increasing the surface water temperature from 0 to 4 °C increases the density of water causing free convection This water circulation transfers heat which in turn augments ice thawing rate and impacts the surrounding rock by heat conduction. The apparent heat capacity method (a temperature-transforming model) and the Darcy approach (a source term methods) are taken to solve the energy and momentum balances. The continuity and momentum Eqs. (1-3) assume an incompressible fluid

controlled by Navier-Stokes equations and can be defined as:

$$\frac{\partial u}{\partial x} + \frac{\partial v}{\partial y} = 0 \tag{1}$$

$$\rho_0 \frac{\partial u}{\partial t} + \rho_0 \left( u \frac{\partial u}{\partial x} + v \frac{\partial u}{\partial y} \right) = -\frac{\partial p}{\partial x} + \mu \left( \frac{\partial^2 u}{\partial x^2} + \frac{\partial^2 u}{\partial y^2} \right) + A \frac{(1-\theta)^2}{\theta^3 + \varepsilon} u \tag{2}$$





$$\rho_0 \frac{\partial v}{\partial t} + \rho_0 \left( u \frac{\partial v}{\partial x} + v \frac{\partial v}{\partial y} \right) = -\frac{\partial p}{\partial y} + \mu \left( \frac{\partial^2 v}{\partial x^2} + \frac{\partial^2 v}{\partial y^2} \right) - \rho_0 g \beta (T - T_0) + A \frac{(1-\theta)^2}{\theta^3 + \varepsilon} v \qquad (3)$$

$u$ and $v$ are velocity components in horizontal and vertical direction, $p$ is the reduced pressure, g is the gravity acceleration, $T$ is temperature and $\mu$ is dynamic viscosity of water. The term $A \frac{(1-\theta)^2}{\theta^3 + \varepsilon}$ imposes velocity transition between solid and liquid. This term is a kind of Darcy-like pressure drop in the fluid momentum equations. When $\theta$ is one (in the presence of liquid phase) this term will be omitted and the full shape of Navier-Stokes equation will be recovered but when $\theta$ is zero, it reduces the velocity to zero to fit the solid phase. $A$ and ε represent arbitrary constants and should be adjusted in each setting. $A$ should be large enough to produce proper damping and allow significant flow in the mushy zone and ε small enough to prevent division by zero in numerical calculation (Mousavi Ajarostaghi et al., 2019). The Boussinesq approximation is used for the buoyancy term. $\beta$ is the thermal expansion coefficient of the liquid phase and is dependent on water density ($\rho_l$) which in turn, is a function of temperature.

$$\beta = -\frac{1}{\rho_l} \frac{d\rho_l}{dT} \qquad (4)$$

$T_0$ and $\rho_0$ are the reference temperature and density at the phase transition. $\theta$ is the fraction of the liquid phase and equals zero when there is only a solid phase and equals to one when there is only the pure liquid phase. It can vary between 0 and 1 in a limited region called the mushy zone, where a combination of both liquid and solid phases is present. This variable depends only on temperature and is important because it determines the interface of liquid and solid phases. It is defined as (Comsol, 2018):

$$\theta = \begin{cases} 0 & T < T_{m1} \\ \dfrac{T - T_{m1}}{T_{m2} - T_{m1}} & T_{m1} \leq T \leq T_{m2} \\ 1 & T > T_{m2} \end{cases} \qquad (5)$$

When the temperature is lower than $T_{m1}$, $\theta$ is zero and only a solid phase is present. On the contrary, when the temperature is higher than $T_{m2}$, there will be only a liquid phase while the mushy zone characterizes a mixture of ice and water and $\theta$ has a value between 0 and 1 depending on the temperature.

The energy equation driving the temperature distribution through the domain is defined as:

$$\rho c_p \frac{\partial T}{\partial t} + \rho c_p \left( u \frac{\partial T}{\partial x} + v \frac{\partial T}{\partial y} \right) - k \left( \frac{\partial^2 T}{\partial x^2} + \frac{\partial^2 T}{\partial y^2} \right) = 0 \qquad (6)$$





$$\rho = \rho_s(1-\theta) + \rho_l\theta \tag{7}$$

$$k = k_s(1-\theta) + k_l\theta \tag{8}$$

$$c_p = \frac{\rho_s c_{p,s}(1-\theta) + \rho_l c_{p,l}\theta}{\rho} + L_m\frac{\partial\alpha_m}{\partial T} \tag{9}$$

$$\alpha_m = \frac{1}{2}\frac{\rho_l\theta - \rho_s(1-\theta)}{\rho_l\theta + \rho_s(1-\theta)} \tag{10}$$


$\rho$, $k$ and $c_p$ are the effective density, thermal conductivity and heat capacity. $L_m$ is the latent heat of melting. $s$ and $l$ subscripts indicate the solid (ice) and liquid (water) phases, respectively. The apparent heat capacity formulation provides an implicit capture of the phase interface by solving for both phases a single heat transfer equation on a fixed grid with effective material properties. The modification of heat capacity incorporates the consideration of latent heat during phase changes specified in

second term of Eq. (9). $\alpha_m$ is an intermediary variable related to the absorption of latent heat during the thawing process. This modification is particularly suitable for materials that exhibit a transition zone near the interface of the phase change, allowing for changes in the topology of the interface. Instead of directly adding a latent heat value ($L_m$) to the energy balance equation at the moment when the material reaches its phase change temperature (in this case, the melting temperature), it is assumed that the transformation takes place within a temperature interval between $T_{m1}$ and $T_{m2}$. It should be mentioned that the mushy

zone does not exist for the perfect pure material and the melting/freezing takes place exactly at melting temperature (0 ℃ for pure water in standard atmosphere). However, dissolution of salt and organic material can increase the impurity of water (Rühaak et al., 2015). In our study, we assumed this interval is between -0.5 ℃ and +0.5 ℃. Nevertheless, the results of the developed model have been tested for other values such as ±0.3 ℃ and ±0.7 ℃ and no significant changes was reported.

Equations (1,2,3 and 6) can be written in dimensionless format (Appendix A). The Rayleigh number reflects the ratio of the

buoyancy force to the viscous force and characterizes the flow regime (laminar or turbulent flow). It is the most important non-dimensional number controlling the thermal behavior of an ice-cleft. Any change in this number yields contrasting results. Conduction is the only mean to exchange heat in the impermeable solid rock surrounding the ice column in a 2D domain. Thermal conduction is obtained by imposing zero velocities ($u = v = 0$) and rock thermal properties ($\rho_r$, $k_r$, $c_{p,r}$) in Eq. (6).

Definition of boundary conditions are depicted in Fig. 1-b. The top boundary is exposed to atmosphere temperature fluctuations

while all other boundaries which are far from the ice-water interface are adiabatic. At the interface between ice-water and rock, hydrodynamic conditions include impermeability and no-slip. The thermal boundary conditions assume temperature continuity and heat flux conservation for this interface. The depth of the ice cleft ($H$) and the aperture size ($A_p$) are 80 and 10 cm, respectively. .

Table_1 shows all the parameters and thermophysical properties of water, ice and rock required for numerical modeling.

Subscripts "s",''l" and "r" refer to ice, water and rock, respectively.



**Table 1. Thermal properties and numerical parameters**

| | |
|---|---|
| $\rho_s$ (kg/m$^3$) | 916.2 |
| $k_s$ (W/m/K) | 2.22 |
| $c_{p,s}$ (J/kg/K) | 2050 |
| $\rho_l$ (kg/m$^3$) | temperature dependent(Guide, 1998) |
| $k_l$ (W/m/K) | temperature dependent(Guide, 1998) |
| $c_{p,l}$ (J/kg/K) | temperature dependent(Guide, 1998) |
| $\mu$ (Pa.s) | temperature dependent(Guide, 1998) |
| $\rho_r$ (kg/m$^3$) | 2320 |
| $k_r$ (W/m/K) | 1.656 |
| $c_{p,r}$ (J/kg/K) | 810 |
| $T_{m1}$ (°C) | -0.5 |
| $T_{m2}$ (°C) | +0.5 |
| $L_m$ (J/kg) | 334000 |
| $A$ (kg/m$^3$/s) | 1000 |
| $\varepsilon$ | 0.001 |

## 2.1 Numerical model

The system of partial differential equations (Eqs. 1, 2, 3 and 6) are solved using the commercial software Comsol Multiphysics

v.6.0. For the mesh sensitivity analysis, four different mesh qualities have been produced for the model geometry in Fig. 1-b. The total number of elements in each of the four cases was 14 k, 24 k, 32 k, and 47 k, respectively. The difference in the results between the latter three cases was found to be insignificant, thus the grid comprising 24 k elements was chosen for the subsequent investigations. The mesh density close to the walls was refined due to the high gradient of velocity and a structured (mapped) mesh type was used for the water-ice domain as shown in Fig. 1-c.






## 3 Model validation

The validity of our model is verified by comparing with studies using two distinct modeling approaches. The first neglects free convection of water while the second introduces water density as a function of temperature and momentum balance (Eqs. (1-3)).

### 3.1 Neglecting free convection

By neglecting free convection, conduction and latent heat fluxes are the only mechanisms transferring heat in water and ice. We compare our results with the results from (Kahraman et al., 1998). These authors developed a model to examine the heat transfer of ice melting inside a 20 cm×20 cm square excluding free convection. A brief description of their conceptual model is given in Fig. 2-a. Ice is at an initial temperature of -30 °C and is exposed to a temperature of 20 °C over half of the lower

boundary ( $0 \le x \le 10$ cm, $y = 0$ cm ) and 70 °C over the other half ( $10 \le x \le 20$ cm , $y = 0$ cm ). The temperature of the solid ice at the top boundary ( $y = 20$ cm) is maintained at 0 °C and the other surfaces (left and right sides) are insulated (Fig. 2-a). Figure 2-b shows the temperature distribution after 5 hours modeled in the present work. Temperature profiles at $x$=4 cm and $x$=16 cm in different times of melting can be seen in Fig. 2-c and d, respectively. The result of our model is in good agreement with (Kahraman et al., 1998).


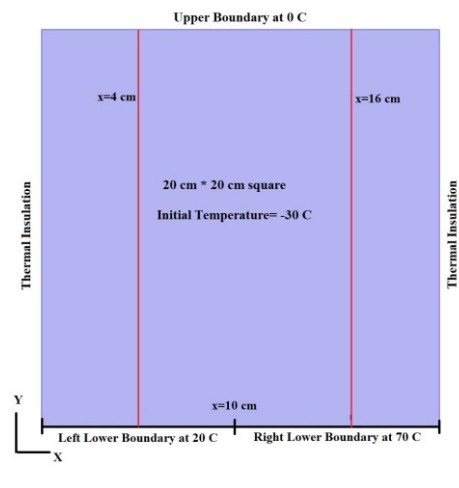

a           b





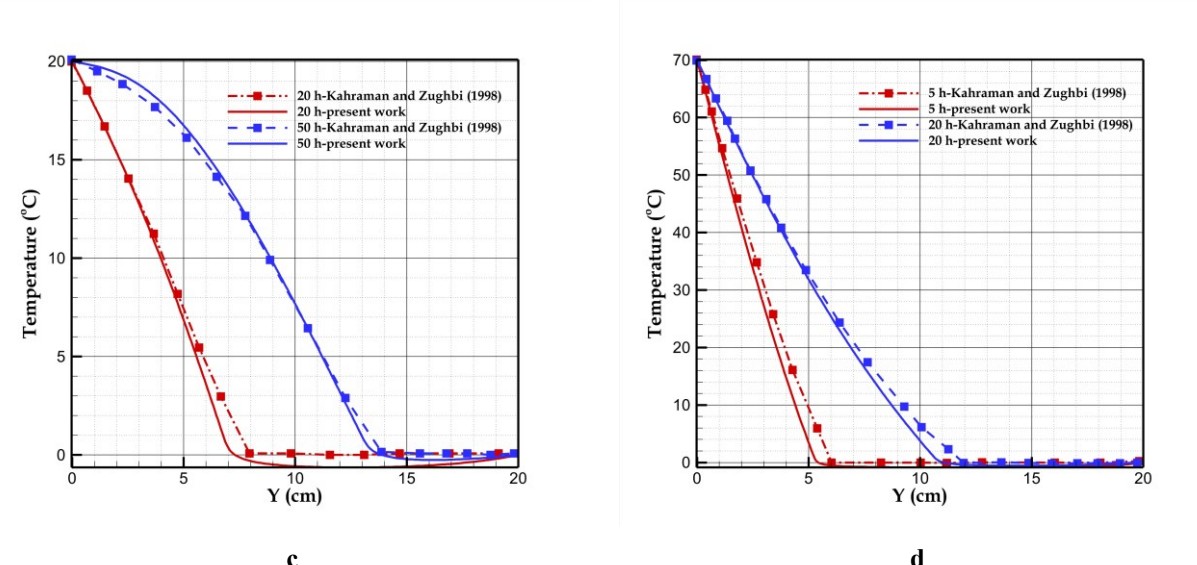

c                                                                                 d

**Figure 2. a) model geometry including initial and boundary conditions b) contour of temperature at t=5 hr c) temperature profile of ice and water at different times of ice melting at *x*=4 cm and d) *x*=16 cm**

## 3.2 Considering free convection

Virag et al., (2006) investigated the effect of free convection on ice melting inside a cavity shown in Fig. 3-a. The lower and
top surfaces are thermally insulated. The left and right boundaries are at 0 and 8 °C, respectively. The density of water increases
between 0 and 4°C, and decreases above 4°C by increasing temperature. After the melting starts, the interface between ice and
water is modified due to a differential melting between the lower and upper parts. Figure 3-b displays contours of temperature
(left) and the velocity field as well as velocity vectors (right) derived by our developed model in this study. The black arrows
indicate the direction of water flow. The model predicts the existence of two free convection cells, a small one in the lower
left corner (when temperature is higher than 4 °C) and a big one in the other part of the liquid region (when temperature is less
than 4 °C). In the lower left cell, the liquid rises upward along the warm wall, as expected in the absence of an anomaly. But
in the other part of the domain the warmer liquid moves downward. The isothermal line corresponding to T=4 °C is plotted
inside the temperature contours implicating well the interface of the two counter rotating convection cells with two temperature
ranges. Furthermore, results show that where the temperature gradient is high, the magnitude of the local velocity increases.
This is seen next to the left boundary and also close to the ice interface. The convective flux homogenizes the temperature in
the water phase due to mixing.





Our numerical model replicates the experimental results (Virag et al., 2006) depicted in Fig. 4 by displaying the ice-water interface at different times. Although some discrepancies exist between the experiments and the numerical model, especially at the bottom of the cavity at the start of the simulation, the overall performance of our model is sufficient to represent the free

convection cells and their effect on ice melting.

**a**

**t=400 s**



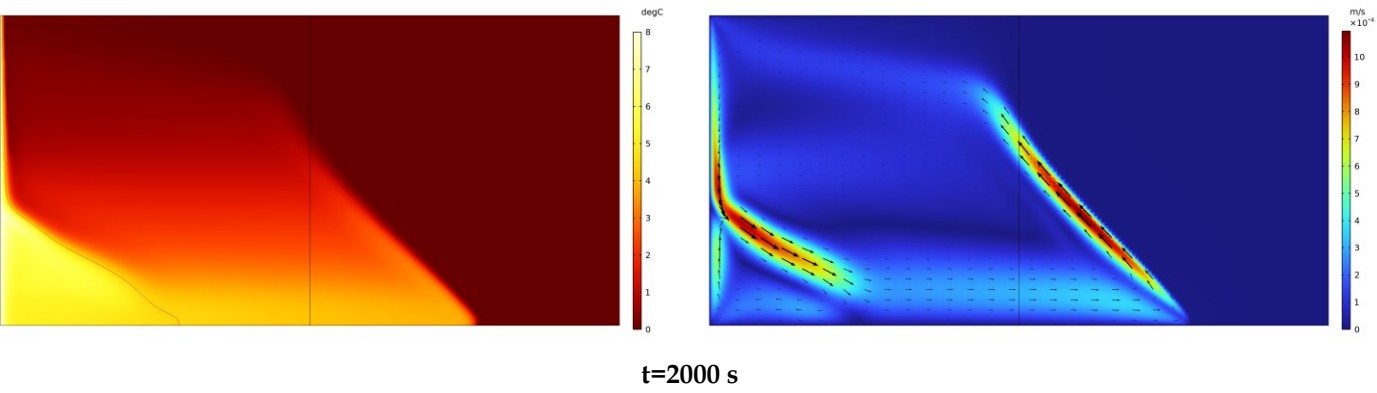

**t=2000 s**

b

**Figure 3. a) Computational domain including boundary and initial conditions in (Virag et al., 2006), b) temperature contours with red isotherm line associated to T=4 °C(left) and velocity magnitude (right) for two different times t=400 and 2000 s**


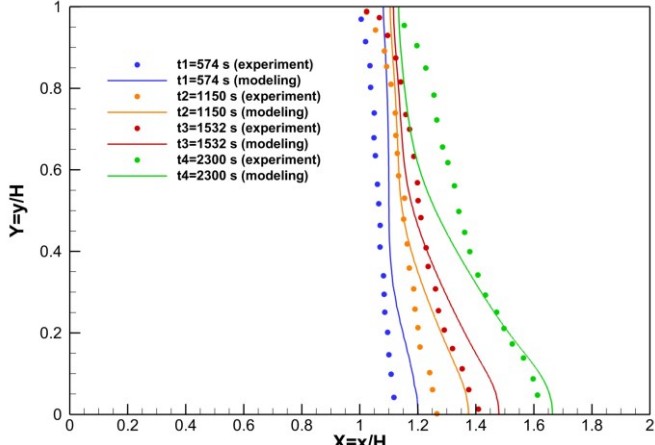

**Figure 4. Evolution of the modeled ice-water interface during melting in comparison with experimental data by (Virag et al., 2006)**





## 4 Results

The effect of free convection in heterogenous, ice-filled karst environment (cf Fig. 1), where the medium is surrounded by rock and prone to atmospheric warming from the top surface was conceptualized in section-2. Here, we investigate the effect

of free convection in more details considering two scenarios under identical thermal settings: 1) "without free convection" and 2) "with free convection" (Fig. 1-b). Figure 5 illustrates the ice fraction for different time steps at 3, 6 and 9 hours for these two scenarios (based on the Fig. 1, the aperture size and the domain height are 10 and 80 cm). The contours illustrate the fraction of ice (1=ice; 0=water). For each specified timestep, the results "without free convection" are shown on the left and the results "with free convection" are displayed on the right. The difference between these two scenarios in terms of melting

rate is obvious. When disregarding convection, a uniform horizontal interface is observed between ice and water and an extreme slope is only observed next to the wall due to the greater thermal diffusivity of rock when compared to water. In fact, the conductive flux thorough the rock propagates faster than in water and melts the lateral side of the ice. In contrast, the presence of convection cells causes a curved interface between the ice and liquid and the total volume of liquid water is noticeably higher compared to the case without free convection.

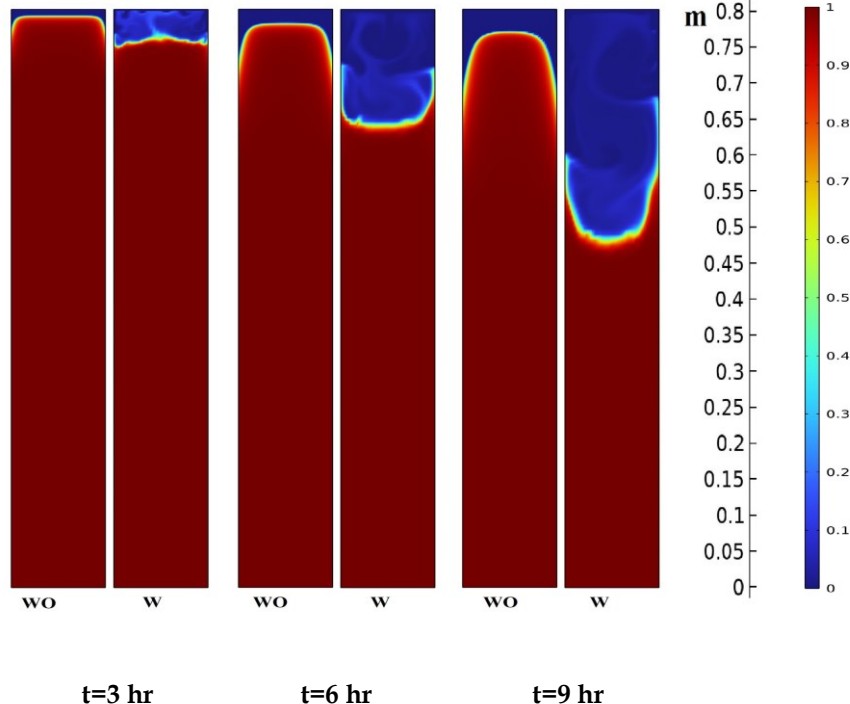

**Figure 5. Ice fraction contour for two different scenarios: without free convection (WO) and with free convection (W) (zero and one in the legend refer to the ratio between the water and ice domains) the aperture size and cavity length are 10 and 80 cm.**

Figure 6 depicts the melting rate for two scenarios considering the absence and presence of free convection in the water phase. Because the initial temperature was -1 °C, the melting starts with c.1h delay. In both cases, the melting rate increases with time




in response to the temperature increase at the top surface and the extending the meltwater depth. After 9 hours, the melting
rate is nearly, one order of magnitude larger when considering free convection (5.1 kg/s) than without free convection (0.6
kg/s). An animation file showing the evolution of the ice fraction can be found in the video supplement of the manuscript
(Sedaghatkish and Luetscher, n.d.).

Figure 7 shows the temperature field of ice, water and surrounding rock after 9 hours by dividing the domain with certain
temperature ranges. The case disregarding free convection displays the conductive vertical temperature gradient (Fig. 7-a) but
when free convection is considered (Fig. 7-b), the water temperature is more uniform between 0 and 1 ℃ occupying most of
the melted part of the cleft. The temperature is higher than 4 ℃ only very close to the walls. In other word, the circulation of
water inside the cleft forms a thermal bridge between ice interface and top atmosphere with a mixing temperature between 0
and 1 ℃. The extent of this mixing zone can be much higher than the diffusion length which is affected efficiently by the
purely conductive flux in the water phase (Fig. 7-a) and is about 6 cm long after 9 hours.

Figure-8 depicts the left wall temperature along the depth in different times of melting for the two scenarios. Although the
melting rate for the case considering free convection in t=3 hr is much higher than the purely conductive scenario, their
corresponding wall temperatures are more and less identical. With more melting, the difference between the scenarios becomes
striking due to the various heat transfer mechanisms taking place.

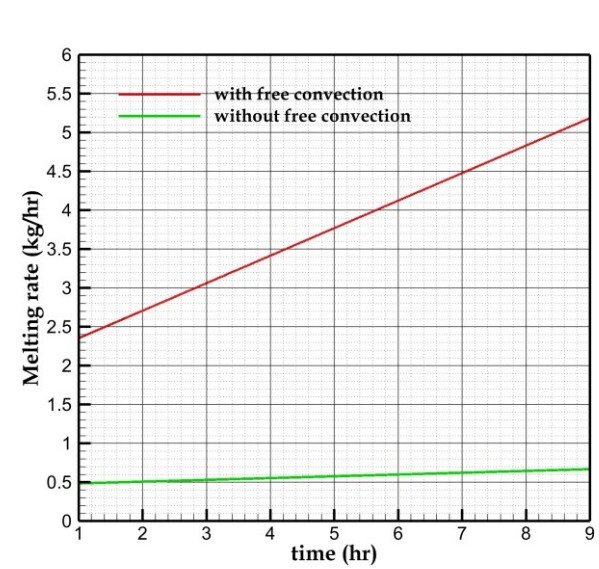

**Figure 6. Melting rate versus time considering two scenarios**



**Figure 7. Temperature contour at t=9 hr for a) without free convection and b) with free convection**





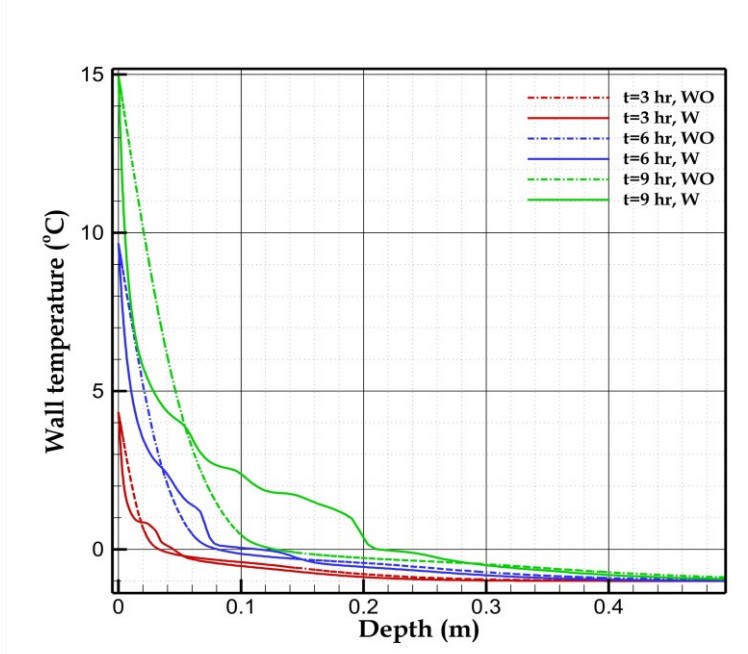

**Figure 8. Left wall (interface of surrounding rock with ice-water filled cleft) temperature as a function of depth from top (atmosphere) to bottom (solid ice)**

Figure 9 illustrates the velocity magnitude as well as the direction of water flow at every point of the ice-water domain over time. Zero velocity should be predicted by the model wherever the water fraction is zero. Irregular water circulations generate a range of local velocities mixing the water and homogenizing the water temperature. As the melting of ice advances and the meltwater amount increases (resulting in a corresponding rise in the Rayleigh number), the velocity of water within the mixing zone increases, thereby accentuating the significance of free convective flux.




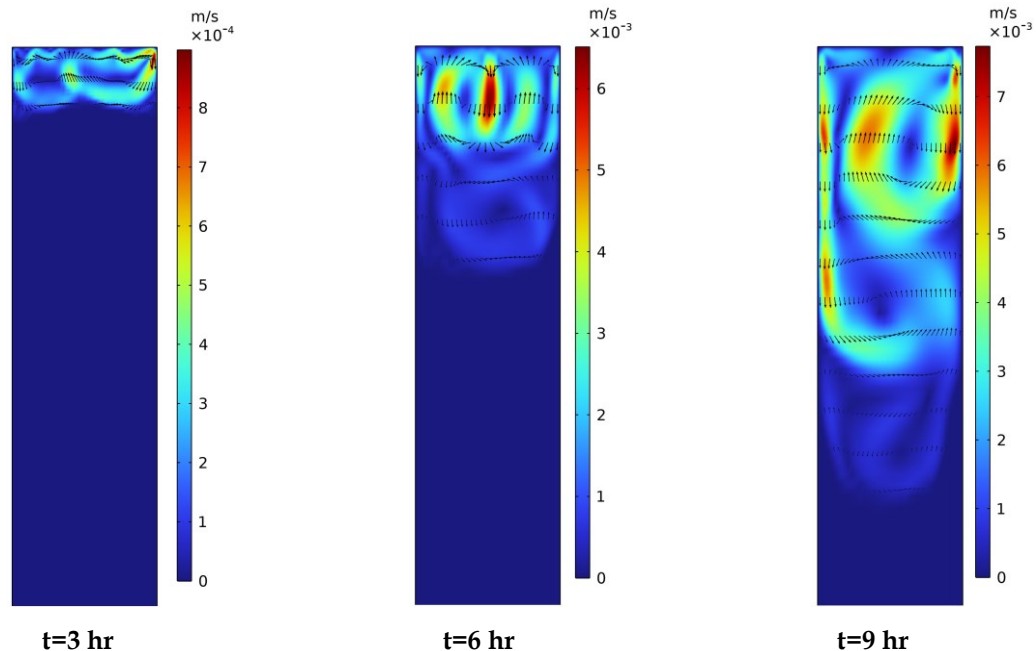

**Figure 9. Velocity magnitude field in the upper 40 cm of the ice-cleft at different times. Zero velocity shows solid ice.**

## 4.1 The effect of aperture size on melting rates

The cleft aperture determines the size of the domain and controls the initial ice mass. We considered different values of aperture size: 2, 10, 20 and 50 cm for doing a parametric study. In order to remove the effect of cross section area associated with

different aperture sizes, we refer to the melting rate per unit of cross section area (kg/hr/m$^2$).

Figure 10 illustrates the effect of the aperture size on the melting rate. For the purely conductive case (indicated by 'WO'), increasing the aperture size to more than 10 cm does not have any effect on the melting rate. But, when the aperture size is smaller than a certain value, the surrounding rock melts the ice more efficiently from the side walls due to its higher thermal diffusivity. In fact, the surrounding rock at a same depth has higher temperature than the ice filled cleft and when the aperture

size is small enough this conductive heat flux propagates perpendicularly to the cleft and melts the ice. In the presence of free convection, the melting rate after 9 hr is almost similar for the aperture sizes 2 and 10 cm (50 kg/hr/m$^2$). For greater aperture sizes, the effect of free convection becomes more significant and the melting rate increases because the effect of surrounding wall completely are removed. It reaches about 110 kg/hr/m$^2$ for 50 cm aperture size.

Figure 10 shows that the melting rate considering free convection with a 2 cm aperture-size (50 kg/hr/m$^2$) is about twice the

rate of the purely conduction case. For higher aperture sizes the difference in melting rates reaches about an order of magnitude!



Decreasing the aperture size reduces the effect of free convection because the viscous forces dominate the buoyancy forces and the fluid eventually becomes stagnant. An empirical correlation provides a rough approximation of an aperture size (or meltwater depth) threshold . Based on Rohsenow et al (1998), when $\frac{A_p}{H} \ll 1$, the following relation is valid:

$$Ra\left(\frac{A_p}{H}\right)^4 \approx 100$$

(11)

Assuming 2 cm aperture size, buoyancy becomes completely negligible at a threshold of 8 m meltwater depth ($H$). Our model
calculates a meltwater depth of 0.3 m after 9 hr meaning that free convection is still affecting the melting rate.

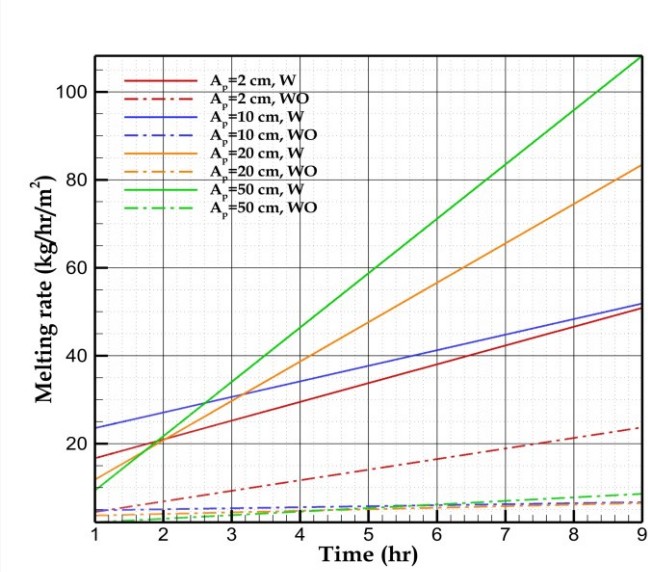

**Figure 10. Effect of the aperture size on the melting rate with (W) and without (WO) free water convection**

### 4.2 Application to field observations

The model is applied to a case-study from Monlesi ice cave, Switzerland (Luetscher et al., 2008). The main purpose is to
qualitatively compare the field data with the model output and test how the pure conduction model can treat these problems. Monlesi ice cave is a low-elevation sporadic mountain permafrost site controlled by a peculiar ventilation regime associated with multiple cave entrances located at a similar elevation (Luetscher et al., 2008). The c. 600 m² cave chamber is partly filled with perennial congelation ice fed by a number of vertical chimneys. Seasonal freezing seals them periodically hindering any further water drainage from the epikarst (Fig. 11-a). The distance between the external surface and the cave ceiling reaches c.



20 m and, within this distance there are ice-filled clefts of different sizes. Daily outside temperature fluctuations thaw the ice and freeze the meltwater in these rock clefts. Figure 11-b displays the atmosphere and the soil temperature at a few centimeters depth during 4.5 days. No rainfall is recorded during this time period and the surface is free of snow. The soil temperature increases with a regular trend between 0 and 5 °C and is noticeably attenuated compared to atmosphere temperature fluctuations. The cave temperature is almost constant around -1 °C implying there is no melting from the bottom of the ice

cleft. Water flow is monitored at one of the main inlets at the cave top wall (the right axis of Fig. 11-b). The daily temperature variation induces a water flow rate with a trend similar to the surface temperature, supporting an origin associated with melting process of ice-filled clefts surrounding the cave.

      To assess the melting rate, we considered the same conceptual model as in Fig. 1 with a linear temperature rise from 0 to 5 °C during 4.5 days at the top boundary (black dash-dotted in Fig. 11-b). In contrast to our model, however, we allow for the

meltwater to drain deeper into the subsurface. The uncertainty about the cleft geometry is very high and we do not know how many fissures feed this water inlet. On the other hand, while our conceptual model is based on a closed cavity for the ease of simulation, the ice-filled cleft can be connected to a network of partially saturated smaller clefts (as in the epikarst) which eventually drains the meltwater to the main vertical inlet at the top wall of Monlesi cave. Eventually, our simplified 2D geometry only approaches the real 3D-environment since other convection cells emerge along the cavity width which cannot

be modeled in 2D.

      A rough estimate of the cavity geometry assumes an initial ice-filled cleft with 3 meters depth and 10 cm aperture size subject to a linear temperature rise at the top boundary (black dash-dotted in Fig. 11-b). The modelled melting rate is in the same order of magnitude as the measured water flow rate (red dash-dotted in Fig. 11-b). A purely conduction-based model underestimates the melting rate by >1 order of magnitude as compared to the case considering free convection (dashed in Fig. 11-b).






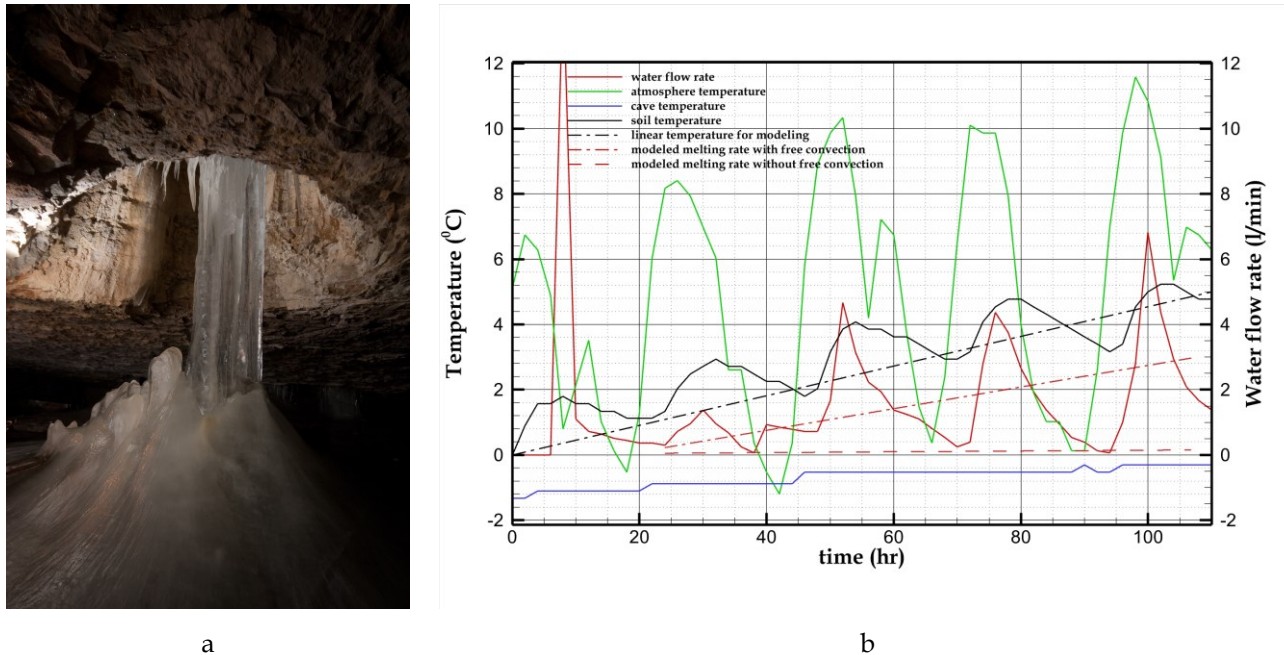

| a | b |

**Figure 11. a) ice filled cleft in Monlesi cave b) field data including atmosphere (green), the soil (black) and cave temperature (blue) as well as the water flow rate measured at the fissure outlet (red) in Monlesi cave. The modelled melting rate considering free convection (red dashed dotted line) is compared to a case including only heat conduction (red dashed lines).**

## 5 Discussion

Field observations from Alpine caves commonly report the presence of ice filled clefts and conduits e.g. (Bartolomé et al., 2022). These clefts thaw seasonally or may remain frozen over longer time-scales to form sporadic mountain permafrost.

Because the cave temperature remains typically below 0°C, the heat has to be supplied from "above". Accordingly, a diphasic medium comprising ice and water fills the rock cleft.

A quantification of the melting rate of ice-rich permafrost in heterogeneous media is essential to assess the speed of permafrost degradation. Here, we show that free convection of water in a sealed cleft enhances the melting rate of interstitial ice dramatically when compared to conventional, purely conduction-based models.

Our model relies on a 2D approach coupling diffusive and convective fluxes in a vertical cleft surrounded by a homogeneous hostrock. The only heat supply is from the upper boundary. Therefore, it does not consider any lateral drainage of air or water in the immediate surrounding of the cleft. Our approach is thus limited to ideal, but nonetheless, plausible case studies where the heat exchange is primarily driven by the surface air temperature. The existence of free convection is inevitable in the same thermal configuration even in different domain geometries such as cylindrical conduits or 3D cavities as long as the Rayleigh



number is sufficiently high based on Eq. (11), even though for a full quantitative assessment, additional simulations based on true geometry are required.

In its present configuration the model assumes that the water at the top boundary is in thermal equilibrium with the hostrock and does not consider the additional advection of sensible heat from seepage of ('warm') water. Whether this water results from the melting of ice in the cleft or recharges from the surface (storm events or snowmelt) does not matter. Free convection

in the liquid phase is triggered only by the physical properties of water, i.e. $\rho_{0°C} < \rho_{4°C}$. The abnormal rise of water-density between 0 and 4°C generates free convection cells as soon as the atmosphere temperature increases. The circulation of water (Fig. 9) functions as a thermal bridge between the atmosphere and the ice, resulting in the formation of a mixed zone with uniform temperature in the water column. In contrast to scenarios involving purely conduction, where the temperature signal from the atmosphere is fully attenuated beyond a certain distance known as the diffusion length, the presence of free convection

allows the uniform temperature to persist over greater distances (Fig. 7-b). This thermal penetration also exerts an influence on the surrounding rock.

Many studies considering thawing in soils assume a broad range of mushy zone for melting/freezing. The water seeping through the soil becomes impure due to overburden pressure, grain size and salt content. This impurity can lead to different freezing temperatures from between -4 °C and -0.1 °C to 0 °C (Tubini et al., 2021); McKenzie et al., 2007; Rühaak et al.,

2015). In carbonate aquifers, the mineralization of the water is directly associated with karst dissolution of the surrounding limestone.

We show that the melting rate of a frozen cleft increases with aperture size when the aperture is larger than a certain value (10 cm). Compared to classical approaches based only on diffusive fluxes we conclude that the consideration of free water convection enhances the melting rate by an order of magnitude when the aperture size is higher than a certain value (based on

Eq. (11)). In the former case, the heat transferred to the cave by conduction is limited to a few centimeters over 9 hours. Despite many uncertainties about the cleft geometry and the measured water flow rate, the modelled melting rate (with free convection) is in the same order of magnitude as the measured water flow rate (Fig. 11) in Monlesi cave.

The effect of free convection is not limited to hourly or daily oscillations and can be studied over much longer timescales, including centennial to millennial fluctuations. Currently, the computational costs are the main barriers for including free

convection in long-term simulations. The full coupling of the momentum and energy equations requires the time steps being much smaller compared to simple conduction-based models. Further investigations are thus ongoing to reformulate the governing equations and simplifying them for simulations over longer time-scales. Moreover, refreezing processes have yet to be considered to fully represent the long-term evolution of such a system.

In this study, we assumed a constant aperture size for modeling ice melt at daily time scale, but it should be noted that in the

long-term, the repeated freeze-thaw cycles can deteriorate the hostrock clefts due to propagation and coalescence of older ones. Maji and Murton (Maji and Murton, 2021) investigated the mechanism and transition of microcracking and macrocracking during freezing and thawing and developed a statistical modeling of crack propagation dynamics based on tension and shearing.



Our results show that, under similar thermal configurations, conduction may not be the principal heat flux and permafrost degradation in heterogenous and ice-rich media may be much faster than anticipated. This is particularly relevant at high-elevation where a clear relationship is observed between rockfall activity and permafrost thawing (Gruber et al., 2004; Ravanel and Deline, 2011; Savi et al., 2021; Morino et al., 2021). In karst systems and fractured aquifers, where secondary porosity is exceptionally well developed, frozen conduits/fractures may all of a sudden drain water into depth and change the local hydrological regime leading a thermal anomaly within the surrounding permafrost (Phillips et al., 2016). The subsequent freeze-thaw processes may lead to the precipitation of so-called cryogenic cave calcites (Žák et al., 2018), a secondary mineral precipitate increasingly used as proxy for paleo-permafrost conditions (Spötl et al., 2021; Töchterle et al., 2023). In contrast to purely conductive systems, our results support a model where rapid hydrological recharge, e.g. due to extreme rainfall events, may accelerate the thawing of mountain permafrost (Luetscher et al., 2013) and thus questions classical interpretations for the formation of cryogenic calcites. But also at shallower depth, acknowledging the potential role of convective heat fluxes in ice-rich permafrost degradation may help predicting the rate of greenhouse gas releases, mainly carbon dioxide and methane, due to the decomposition of formerly frozen organic matter (Schaefer et al., 2014; (Schuur et al., 2015). Eventually, the effect of water free convection on ice melting rate is not limited to permafrost regions. For instance, the melting of icebergs (kilometer-scale floating glacial ice) can also be impacted by water free convection (Couston et al., 2021; Hester et al., 2021) increasing production of freshwater in oceans with potential impacts on the climate at global scale.

## 6 Conclusion

The main conclusions of this study can be summarized as follows:

In heterogeneous and ice-rich media, the melting rate is noticeably controlled by the free convection of water. A model considering only conduction may underestimate melting rates of frozen rock clefts by one order of magnitude.

Free convection gives rise to a mixing zone characterized by a uniform temperature. This zone has the potential to extend beyond the length typically associated with diffusion. Importantly, the presence of this mixing zone does not only impact on the temperature distribution within the water, but also affects the surrounding rock.

The effect of free convection vanishes with decreasing aperture size, typically < 2 cm at 8 m meltwater depth, when the viscous force at the wall dominates buoyancy forces. In this case, the meltwater cannot flow and conduction is the main heat flux responsible for the melting rate.





## Appendix A: Dimensionless form of governing equations

All the governing equations including Eqs. (1, 2, 3 and 6) are written in non-dimensional format to identify the important and controlling non-dimensional numbers. This helps the analysis of different problems under the same hydro-thermal setting. The dimensionless variables for length, time, velocities, pressure and temperature can be defined as:

$$x^* = \frac{x}{H}, y^* = \frac{y}{H} \tag{A1}$$

$$t^* = \frac{t\alpha}{H^2} \tag{A2}$$

$$u^* = \frac{uH}{\alpha}, v^* = \frac{vH}{\alpha} \tag{A3}$$

$$p^* = \frac{pH^2}{\rho_0 \alpha^2} \tag{A4}$$

$$T^* = \frac{T - T_c}{T_H - T_c} \tag{A5}$$


$\alpha$ is thermal diffusivity, H is the meltwater depth varying during the ice thawing (its maximum value is equal to height of domain). Here, $T_h$ can be regarded as the atmosphere temperature at the upper boundary and $T_c$ is the ice interface or melting temperature. After substituting all the variables, the non-dimensional form of equations can be written as:

$$\frac{\partial u^*}{\partial x^*} + \frac{\partial v^*}{\partial y^*} = 0 \tag{A6}$$

$$\frac{\partial u^*}{\partial t^*} + u^* \frac{\partial u^*}{\partial x^*} + v^* \frac{\partial u^*}{\partial y^*} = -\frac{\partial p^*}{\partial x^*} + \boldsymbol{Pr}(\frac{\partial^2 u^*}{\partial x^{*2}} + \frac{\partial^2 u}{\partial y^{*2}}) + \boldsymbol{C}\frac{(1-\theta)^2}{\theta^3 + \varepsilon} u^* \tag{A7}$$

$$\frac{\partial v^*}{\partial t^*} + u^* \frac{\partial v^*}{\partial x^*} + v^* \frac{\partial v^*}{\partial y^*} = -\frac{\partial p^*}{\partial y^*} + \boldsymbol{Pr}(\frac{\partial^2 v^*}{\partial x^{*2}} + \frac{\partial^2 v^*}{\partial y^{*2}}) - \frac{\boldsymbol{Ra}}{\boldsymbol{Pr}} T^* + \boldsymbol{C}\frac{(1-\theta)^2}{\theta^3 + \varepsilon} v^* \tag{A8}$$

$$\frac{\partial T^*}{\partial t^*} + u^* \frac{\partial T^*}{\partial x^*} + v^* \frac{\partial T^*}{\partial y^*} - (\frac{\partial^2 T^*}{\partial x^{*2}} + \frac{\partial^2 T^*}{\partial y^{*2}}) = 0 \tag{A9}$$

Rayleigh (*Ra*) and Prandtl (*Pr*) numbers (A10 and A11) are the main non-dimensional numbers which control the behavior of the system ( $\upsilon$ is the kinematic viscosity of water). The Rayleigh number is associated to ratio of buoyancy force to viscous



force and characterizes the flow regime (laminar or turbulent flow). In this study, its value changes with the melting evolution ($T_h$ and H are increasing with time due to increasing atmosphere temperature raising on top surface) but the maximum possible Rayleigh number yielding laminar flow is about $10^9$. The Prandtl number is the ratio of momentum diffusivity to thermal

diffusivity and is mainly related to the physical properties of the fluid. In this study, $Pr$ is approximately constant ($Pr$=13). The other non-dimensional number is $C$ (A12) which forces a zero velocity in ice and does not have any physical meaning.

$$Ra = \frac{g\beta(T_H - T_c)H^3}{\alpha\upsilon} \tag{A10}$$

$$Pr = \frac{\upsilon}{\alpha} \tag{A11}$$

$$C = \frac{A H^2}{\rho_0 \alpha} \tag{A12}$$

*Software availability*. The Comsol file (closed source software) is available upon request.

*Video supplement*. The animations of ice fraction evolution are available at https://doi.org/10.5281/zenodo.8435168.

*Author contributions*. AS completed the modelling work and wrote the original manuscript. ML designed the study together

with AS and contributed with field data from Monlesi ice cave. All authors discussed the results and contributed to the final edition of the paper.

*Competing interests*. The authors declare that they have no competing interests.


*Acknowledgements*. This project is supported by the Swiss National Science Foundation (project n°200021_188636).



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
