# Peer review of "Modeling the effect of free convection on permafrost melting-rates in frozen rock-clefts"

_EGUsphere, 2023_

## Author Comment (AC1)

**General comments**

In their study, Sedaghatkish et al. model ice melt in ice cleft and compare melting rates with or without free convection. They show that free convection is a key process to account for because it increase melt rates by an order of magnitude. The modeling works is based on a commercial software, validated against to modeling experiments from the literature as well as observations. When used in a real world case, their setup shows a good ability to produce a melt rate that matches the water flow observed in a monitored cave.

My overall impression is that the study tackles an interesting topic with a relevant angle and adequate methodology. The model validation is convincing and the results on the real world case are good. The article is well structured and reads well. Some illustrations could be prettier but do the job (see suggestions below). Overall I think it is a good study that maybe won't catch the attention of everyone because it tackles a very specific question but that TC should be interested in publishing because it is a relevant piece of work on the cryosphere. I don't see much to change, so I suggest minor revision to the editor.

I made a detailed lists of small comments, I think some significant progresses can be made on the clarity of the explanations. The biggest of my small comments are the following.

- I think the study can be more clear and pedagogic regarding the physics and the equations.

- I think the author should discuss to what extent, fast water flow through the macro-porosity of this kind of massif could compete with free convection by disturbing the small scale density contrast that it requires. Who wins? Would it be nice to have a model that do both… (see below)

- I think the discussion should also try to discuss larger scale implications of the results in terms of consequence for permafrost disappearance at the scale of the massifs/mountains and regarding catchments water balance.

We thank the reviewer for the constructive comments, particularly related to the governing equations and discussion section. We are delighted to hear that our study is acceptable for The Cryosphere journal with minor revisions. We tried our best to modify or justify better our statements and claims in order to boost the quality of the manuscript. We have updated the methodology section by adding more details and made a better connection between our study and other fields by considering the effect of free convection also at larger scales. A list of references used in our answers is available at the end of the document.

Abstract: I have the feeling that it could be nice to remind in a few words what is free convection to the readers in the abstract as it took me a few minutes to realize what we are talking about. E.g. "free convection (convection driven by density contrasts within the water phase)". Instinctively I did not think that the water bodies in karstic environments are big enough for free convection to be important, like it would be in a lake for example.

The abstract will be modified as follows:

*"**Abstract**. This research develops a conceptual model of a karst system subject to mountain permafrost. The transient thermal response of a frozen rock-cleft after the rise of the atmosphere temperature above the melting temperature of water is investigated by numerical simulations. Free convection in liquid water (i.e., buoyancy-driven flow) is considered. The density increase of water from 0 to 4°C causes warmer meltwater to flow downwards and colder upwards, resulting in*

*significant enhancement of the heat transferred from the ground surface to the melting front. Free convection increases the melting rate by approximately an order of magnitude compared to a model based on thermal conduction in stagnant water. The model outcomes are compared qualitatively with field data from Monlesi ice cave (Switzerland) and confirm the agreement between real-world observations and the proposed model when free convection is considered. »*

**L15: "the anomalous behavior of water between 0 and 4°C"Do you refer to the density increase of water from 0 to 4°C? If so state it in a less mysterious way.**

The density increase of water between 0 and 4°C is better explained in the new version. The following paragraph has been inserted along with a new figure showing the water density as a function of temperature (see below).

*"Figure 1-a displays the bottom of a frozen cleft hosted in an Alpine karst. Such clefts and fissures are characterized by distinct geometries accommodating contrasted volumes of ice. Our aim is to study the effect of free convection of water on the melting rate in frozen rock clefts and/or karst conduits at daily scale. Atmospheric warming at the upper boundary melts the ice from top of the fractures, and increases the meltwater temperature. While most fluids expand as temperature increases, water shrinks when warmed from 0°C to 4°C. Above this temperature, the common behavior is recovered (see the maximum density at 4°C in Fig.2). Therefore, the progressive warming of the meltwater at the top of the cleft results in an unstable situation (heavier fluid above lighter) that triggers free convection (buoyancy-driven flow). »*

[Figure]

**Figure 2: density of liquid water as a function of temperature.**

**L22: The first sentence talks about the impact of global climate change on permafrost and the ref is only about the French Alps, maybe add a more large scale ref as well.**

We added some other references (Walvoord and Kurylyk, 2016; Jin et al., 2021) relating to larger scales. The first one is a review paper about climate warming and its consequences on soil thermal change which are expected to modify the distribution of permafrost, leading to changing hydrologic conditions, including alterations in soil moisture, connectivity of inland waters, streamflow seasonality. The second one is another review paper which is about permafrost degradation effect on arctic and alpine ecology and vegetation.

*"With global climate change and rising temperatures, permafrost degradation has become a significant concern (Duvillard et al., 2021; Walvoord and Kurylyk, 2016; Jin et al., 2021)"*

**L33-34: "have shown that heat advected by water and air fluxes may significantly disturb the geothermal field, challenging classical models of heat propagation based on conductive fluxes"**

**Here you want to talk about the general case, not specifically karstic environment, so it would be good to come up with a reference that demonstrates that in non-karstic environments. Since you then talk about well-developed conduits right after, it would be also nice to add a line to explain the difference between convection in a porous media and convection in a conduit.**

The references to the boreholes do not specifically refer to karstic environments. Nonetheless, we added. In contrast to porous media, karst systems concentrate water fluxes through well-defined conduits. It is rare to have similar fluxes in porous media. We do not think this statement needs to be underlined further.

**L36: "this permafrost" which permafrost? Maybe better "permafrost in karstic environments" or something similar. But the "this" refers to something undefined I believe.**

Corrected.

*"The discontinuous nature of permafrost in karst environments may lead to the formation of massive cave ice at depth (Bartolomé et al., 2022) but also explain unexpected speleothem formations during glacial times (Luetscher et al., 2015; Fohlmeister et al., 2023; Fohlmeister et al., 2023)"*

**L42: If rock-glaciers are relevant to this list, add it.**

Added.

**L43: lower than what? Than 2 or 3D models?**

Corrected.

*"they offer several advantages including lower computational costs than 2or 3D models and easy implementation"*

**L47: "This distributed model is efficient for large domains (catchment scale)."**

**In what regard? What does that mean?**

We now specify *"for calculation of temperature in large domains".*

**L49-50: Tubini et al., (2021)With which processes? Conduction only or also convection?**

We now specify "based on conduction and latent heat flux".

*"Tubini et al., (2021) proposed a numerical approach for modeling heat transfer of permafrost thawing based on conduction and latent heat flux in 1D domain which is capable to deal with high time steps and maintains conservation of energy in long-term simulations"*

**L53-54: "The effect of water flow inside the clefts is noticeable because of creating thermal shortcut between atmosphere and subsurface."This is a general statement that does not really fit the list of modelling studies you are going through. It probably fits better earlier when you talk about medias with conduits.**

The statement refers to the model of Hasler et al., (2011) mentioned in the previous sentence. This has been clarified as follows:

*"Hasler et al., (2011) investigated the effect of advective heat transport in frozen rock clefts and fractures at small scales. They built a conceptual model combining numerical modeling and laboratory experiments. These authors find a significant effect of water flow inside the clefts, due to the formation of thermal shortcut between atmosphere and subsurface."*

**L59-60: Maybe one more sentence to be more specific. You imply that atmospheric warming could warm melt water located close to the upper boundary close to 4°C, which would later sink right?**

See our answer to L15 above.

**Figure 1. As I am still at the point where I try to understand what you did, I am surprised that you show on the picture a volume which order of magnitude is tens of meters and your modeled domain is in the order of 80 cm. I missed earlier on some explanations on why 80 cm is a relevant size and how something that small can have a larger scale relevance (i.e. small features but very frequent I suppose).**

Thank you for your feedback. Indeed, we realize that the description of the problem was confused. Section 2 will be completely rewritten. It will begin by a section 2.1 describing the computational domain with the main physical assumptions:

*"2.1 Physical model and simplifying assumptions*

*We consider the upper part of a single vertical cleft of size aperture $A_p$ filled with pure water whose melting point is $T_m = 0°C$. This cleft is surrounded by a rock mass of width W (see Fig.1b). In karst massifs, water flow concentrates in well-defined conduits (Ford and Williams, 2007). The micro-porosity of the rock is thus disregarded, and impermeable rock mass is assumed.*

*The cleft is located at the center of the 2D domain of height $H_{dom}$. In the initial state, the system (water and surrounding rock) is at the uniform initial temperature $T_i$=-1°C, and all the water is frozen. At time t=0, the temperature of the ground surface $T_s$ increases at the constant rate 1.77 °C/hour to reach 15°C after 9 hours. This temperature increase is similar to the daily warming between the early morning and the afternoon.*

*The effect of the cleft aperture size was investigated by varying $A_p$ from 2 cm to 50 cm. We imposed $H_{dom}$=0.8 m and W =1 m in all simulations. These values are large enough so that the thermal perturbation induced by the presence of the cleft does not reach the domain boundaries at the end of*

*the simulated time (9 hours). The vertical and bottom boundaries of the domain can therefore be considered as adiabatic (see Fig.1b). It is important to note that the domain height $H_{dom}$ contains only the upper part of the cleft, whose actual depth commonly ranges from 1 to 10 m. The value of $H_{dom}$ used in this study is convenient for the daily time scale considered in the numerical simulations. Simulating larger time scales would require larger values of $H_{dom}$."*

**Also Figure 1b can be improved. Remind what are the x and y axis (I am surprised you did not use z for the vertical axe by the way, I have the feeling it is what the general reader would expect to understand your work more smoothly). As it is I am still unsure which one is the vertical one. If you put dT/dx on the side I am tempted to believe it is a boundary condition for the side, but with the proximity of the other dt/dx to the H, I would be tempted to think x is the vertical axis. Add arrows as well maybe. The red arrows look like they were made with MS Paint. I think in general you can give more love in Fig 1B.**

A new figure 1-b has been inserted (see below). The vertical axis will be called *z* instead of *y* throughout the manuscript. x and z will be explicitly defined in the governing equations section.

[Figure]

**Figure 1: b) Computational domain with external boundary conditions; 2 cm $\leq A_p \leq$ 50 cm, $H_{dom}$=0.8 m and $W_{dom}$=1m (the sketch is not at scale).**

**L86: The sentence misses a point at the end.**

Corrected.

**L93: I feel there is maybe a lack of pedagogy regarding the A x… term that is introduced. I don't know what is a Darcy like pressure drop (a bit of physics "with the hand" to help intuition maybe) and I don't see why theta=0 will nullify the velocity. What I see is that you will add the term A v /**

**epsilon to the Navier Stokes equation (or A u / epsilon horizontally). How does that nullifies the velocity?**

**Similarly, I think you should explicit what is the Boussinesq approximation. I don't suspect many readers of TC will know that and it can be disorienting to see rho0 in your equations whereas you intend to work with density contrasts in your study (and your actual rho variable is hidden in beta, so at first glance, rho seems to be fixed).**

We used a method that allows to define a single set of governing equations in both solid and liquid phases of water. We realize that the presentation of this method was confused. It will be rewritten in the future version, focusing on the main principles, and referring the reader to the literature for more detailed explanations.

The Boussinesq approximation consists in assuming constant density in the Navier-Stokes equation except in the buoyancy term. This is a standard approximation valid most of the time in liquids. In the new paper version, it will be defined and justified.

**L109: "domain", to me at this stage, it is not clear if your domain is just the water/ice or also the surrounding rock. I see a mesh on the rock on Figure 1 but the rock is impermeable. So I think you should find a way to be clear on this, talking about the "water and ice domain only" or the "whole domain" or anything that would reach the same goal. So for Temperature, your domain is water/ice + rock? Because the cp you describe looks like it is for water only, it is not a freeze curve that would account for suction effect in the rock, that would spread the phase change below 0°C.**

See the new section 2.1 above and new figure 1b. We hope that the new text of section 2 will be more clear.

**L131: "no-slip" same, explicit quickly. Also what about heat fluxes between the water and the wall? Is it just conduction or also convection?**

The following clarification will be inserted in the section dedicated to the governing equations:

*"The boundary conditions are as follows. At the interface between an impermeable solid and a viscous fluid, the fluid velocity is equal to that of the solid (see Guyon et al (2015)). This is the so-called no-slip and impermeability conditions, resulting in $u = v = 0$ at the rock-water interface. The temperature continuity and the heat flux conservation through this interface are also considered (since the water velocity vanishes at the rock-water interface, the heat flux through the interface reduces to conduction). As already mentioned in section 2.1, the bottom and vertical external boundaries are adiabatic, and the temperature evolution of the top boundary is imposed (see Fig.1b)."*

The following reference will be added: « Physical hydrodynamics, E. Guyon, J.P. Hulin, L. Petit, C.D. Mitescu, Oxford University Press, 2015 »

**Section 2.1. Finite elements? Finite volumes?**

The following sentence will be inserted in the new version:

*"The system of partial differential equations (Eqs. 1-5) was solved by finite elements using the software Comsol Multiphysics version 6.0 (Galerkin method with quadratic Lagrangian elements, time discretization by implicit backward differentiation formula)."*

**L151-152 : Here you just mention comparison with simulations even though Virag also has observation if I understood correctly. Reproducing observations is an even better validation, so make it more clear.**

I think you are referring to Kahraman et al. We are comparing our result with the modeling results of Kahraman et al when free convection is absent. In fact, in their study, they investigated also the effect of free convection both experimentally and numerically. However, we prefer as a first step to consider the purely conduction scenario as the simplest possible test case before considering a more intricate case including free convection. The introduction to section 3 was clarified as follows:

*"The validity of our model is tested by comparison with two studies from the literature. A simple test case assuming stagnant liquid water (no free convection) was selected as a first step (numerical simulation of ice freezing by Kahraman et al., (1998). In a second step, our model was tested against experimental results including free convection (ice melting experiment by Virag et al., 2006). "*

**L159: « with (Kahraman et al., 1998). »Fix parenthesis**

corrected.

**L172-174 « The isothermal line corresponding to T=4 ℃ is plotted inside the temperature contours implicating well the interface of the two counter rotating convection cells with two temperature ranges. »**

**I do not understand this sentence. Please reformulate.**

In fact, the isothermal line shown in figure 3-a is the interface of two convection cells as shown in figure 3-b. The velocity direction of these two convection cells are opposite. We added some details for better understanding:

*"The 4°C isotherm is plotted together with the temperature contours underlining the interface of the two convection cells rotating in opposite directions and showing two distinct temperature ranges (Fig. 3-b)."*

**Sect 3-2**

**Here be more precise whether you replicate the simulations from Virag or their observations.**

We modified the text:

*"Our numerical model replicates the observations of Virag et al (2006) depicted in Fig. 4 by displaying the ice-water interface at different times"*

**L165: "The density of water increases between 0 and 4°C, and decreases above 4°C by increasing temperature."If not already the case, this should appear earlier. Not in the experiment explanation.**

This sentence has been removed, and more explanation given in the introduction about the density maximum of water.

**L196-197 "thorough"Through**

Corrected.

**L200 "for two scenarios"**

**They are the same 2 scenarios as *before* right?**

Yes. It was modified for better clarification:

*"Figure 6 depicts the melting rate for both previous scenarios considering stagnant water of free convection in the water phase"*

**L202: "and the extending the meltwater depth"Problem with the sentence.**

The sentence was modified as follows: "In both cases, the melting rate and the meltwater depth increase with time in response to the temperature increase at the top surface"

**Figure 8. Make a more explicit legend than W/WO.**

In the new version, we will use the abbreviations SLW for stagnant liquid water and FC for free convection. They will be defined in the introduction and recalled in figure captions.

**L221: "Irregular water circulations"What is that?**

Deleted "Irregular".

**L238: "completely are removed."**

corrected into: *"the effect of surrounding rock can be disregarded".*

**L240: You left an exclamation mark.**

Deleted.

**L242-L245: not completely clear to me, can you reformulate? What is a aperture size threshold? So you back calculate Ra based on the empirical relationship and check for threshold values? With the last sentence do you mean that the empirical relationship is not valid?**

This section has been rewritten as follows:

*"In the present work, we simulated 9 hours of atmosphere temperature increase. When the aperture size $A_p$ was varied from 2 to 50 cm, the liquid height H at the end of the simulation approximately ranged from 30 to 40 cm, and the convection cell occupied the entire liquid domain. However, the liquid height reached after 9 hours is only a small part of the actual height of the cleft (commonly up to 10 m). H is expected to increase if longer times are considered. The question arises whether the*

*free convection cells always fill the entire liquid domain at longer times, despite the increase of friction due to lower aspect ratio $A_p/H$. If the convection cell occupies only a part of the cavity, the efficiency of heat transfer between the ground surface and the melting front will be reduced. The significance of free convection can be assessed from the value of the dimensionless Rayleigh number*

$$Ra = \frac{g\beta(T_c - T_H)H^3}{\alpha_l \, \nu_l} \qquad (10)$$

*where $(T_c\text{-}T_H)$ is the temperature difference between bottom and top surfaces, $\alpha_l$ and $\nu_l$ are the liquid water diffusivity and kinematic viscosity, respectively. Ra represents the ratio of the diffusion time over the free convection time ($Ra \sim 10^8$ in the numerical experiments presented in this article). In a cavity with infinite lateral dimensions, free convection is triggered when $Ra \gtrsim 10^3$ (otherwise, the conductive state is stable, see Bergman et al (2017) for more information about the Rayleigh-Bénard instability). However, in the confined geometry considered in this work, the presence of the vertical walls must be considered. Rohsenow et al (1998) provide the following condition for convection onset, which takes into account the stabilizing effect of the vertical walls for $A_p \ll H$, in the limiting case of perfectly conducting walls:*

$$Ra \gtrsim 10^2 \times \left(\frac{H}{A_p}\right)^4 \qquad (11)$$

*Injecting Eq.(10) in Eq.(11) yields the maximum value of the liquid height H for which the free convection cell extends from the ground surface to the melting front:*

$$H \lesssim 10^{-2} \times \frac{g\beta(T_c - T_H)A_p^{\,4}}{\alpha_l \, \nu_l} \qquad (12)$$

*Considering that the liquid region at temperature T>4°C is stable and that the isotherm 4°C is close to the top of the cleft when the free convection cell fills the entire cavity (see Fig. 7b), we get $(T_c\text{-}T_H)$=-4°C. Using the physical properties from section 2.3 and $A_p = 2$ cm (the minimum aperture size considered in this study) yields $H \lesssim 10$ m, which is also the order of magnitude of the maximum cleft height. Therefore, free convection cells should always extend throughout the melted region for $A_p \gtrsim 2$ cm. Note that the assumption of perfectly conducting walls used in Eq.(11) is less favorable to convection than finite conductivity (Rohsenow et al, 1998). Eq.(12) is thus expected to slightly underestimate the higher bound of H corresponding to fully developed free convection cells."*

**L257: What is the soil temperature? Do you have a real pedological soil? To make things clear maybe you can talk about the ground surface (I guess it's the ground surface outside right, not in the cave?).**

The soil temperature is given in Figure 11 and was measured at 10 cm depth in a true pedological soil (calcisol – B horizon).

**L260-261: "The daily temperature variation induces a water…"Temperature of what? A cave is a complex setup compared to an homogenous media, be more specific to help the reader get a clear picture of what you describe.**

Corrected as follows:

*"The atmosphere daily temperature variation induces a water flow rate (0-12 l/min) with a trend similar to the surface temperature, supporting an origin associated with melting process of ice-filled clefts surrounding the cave."*

**L264: "we allow" you allow but you are talking about the natural case no? Because you said "In contrast to our model" just before?**

Thank you for your attention. The text was corrected: "In contrast to our model, meltwater drains deeper into the subsurface".

**L273: "(red dash-dotted in Fig. 11-b)"If I understand correctly, should be "(red solid line and red dash-dotted line…" because the good news is that the dash-dotted line follows the moving average of the red solid line right? I feel you should make the 3 red lines thicker or anyhow more visible. They convey the key message of your paper right?**

You are right. We changed them to thicker lines to make them more visible:

[Figure]

**Discussion : the 2 first paragraphs feel redundant with things you already said in the introduction and methods. I would start at line 285.**

We feel it is important to remind the context for the discussion but agree it can be simplified to avoid to many overlaps. Accordingly, we deleted the first paragraph and now start with "A quantification of the melting rate …"

**L293-295: "Whether this water results from the melting of ice in the cleft or recharges from the surface (storm events or snowmelt) does not matter."I don't understand what you mean with this sentence. Additionally, if these are the sources of water, it is unlikely to be warm, so I miss the connection with the previous sentence.**

The key element controlling the melting rate is the presence of a liquid phase which may result from the melting ice column or from hydrological recharge. Accordingly, permafrost thawing is also enhanced by rainfalls (and not only temperature). We agree that the formulation of this paragraph was unclear. It was clarified and simplified in the next version of the article.

**L302-306: I don't see where this paragraph goes. In soils things work differently than in karstic cavities right? The freezing will spread below 0 because of the suction in the soil. I don't see what perspective that gives on your work.**

Indeed, the effect of suction in soils is not relevant for the freezing of a cleft. This paragraph will be removed. Instead, we will add in the conclusion that the mineralization due to karst dissolution should be considered in future works (expected effects are a shift of the melting temperature and also a contribution of salt concentration gradient to buoyancy).

**L312 :"… is in the same order of magnitude as the measured water flow rate (Fig. 11) in Monlesi cave." Here would be a nice place to discuss why the water flow is oscillating and your melting rate is smooth. If the average values are similar, what create the contrast in timing?**

The measured water flow rate has variable amplitudes in its daily fluctuations which can be related to uncertainties in the epikarst flow paths and the meltwater saturation within the clefts. Here, our goal is to compare two different scenarios (with and without free convection) under the same thermal configuration and demonstrate that purely conduction case (without free convection) is too far from the measured water flow rate even though the "with free convection" scenario may not be too close to our observations.

We imposed a linearized soil temperature indicated by the black dash-dotted line in figure-11 at the top boundary in our model. Accordingly, the corresponding melting rate is linear too as expected but in the same order of magnitude with water flow.

**L313: "The effect of free convection is not limited to hourly or daily oscillations and can be studied over much longer timescales, including centennial to millennial fluctuations. "When melting very big ice clefts? If the weather warms up from one year to another, once you start melting a cleft of one or 2 meters long, isn't it going to melt in a few years? Or are you discussing the melt at the scale of a massif?**

Some ice-cleft can reach about 50 m length with a few meters aperture size. The effect of climate warming can be seen on the long-term. It is of interest to characterize the melting rate and/or the long-term changes in the meltwater depth due to freeze/thaw cycles. Further investigations are required in order to developing a model with feasible computational time.

**L326-328: "In karst systems and fractured aquifers, where secondary porosity is exceptionally well developed, frozen conduits/fractures may all of a sudden drain water into depth and change the local hydrological regime leading a thermal anomaly within the surrounding permafrost (Phillips et al., 2016)."That's something you could discuss further to think against yourself. How much fast flow in karstic system is likely to actually advect temperature quickly over long distances and disrupt the peace of free convection? It gives an opportunity to discuss the representativity of the process you pinpoint in the perspective of the general functioning of a karstic/fractured massif exposed to seasonal/long term cold weather. It Is also an opportunity to compare your work with Hasler et al. (2011). Is there a process more important than the other? Would we gain something trying to represent both at the same time?...**

The model proposed by Hasler et al. (2011) relies on experimental data, which makes difficult a direct comparison with our model. However, water drainage can clearly enter in competition with free convection. Determining what mechanism dominates is an interesting question that will be mentioned in conclusion as an outlook for future works.

**L333-335 "But also at shallower depth, acknowledging the potential role of convective heat fluxes in ice-rich permafrost degradation may help predicting the rate of greenhouse gas releases, mainly carbon dioxide and methane, due to the decomposition of formerly frozen organic matter (Schaefer et al., 2014; (Schuur et al., 2015)."This sounds a bit far-fetched. What carbon pools are you talking about? There is not much carbon in the fractures/karstic cavities of a rock massif right? And you do not expect much free convection in an organic peat soil right? I have the feeling that free convection is relatively low in the list of missing processes to accurately represent permafrost thaw where you find a lot of organic carbon, but I am happy to be proved wrong. You can check Kane et al. (2001, GPC, 10.1016/S0921-8181(01)00095-9).**

Thank you for your pointing this out. We believe any kind of ice-rich media (not only fractures/karst cavities) can be subject to the effect of free convection. Global warming increases the top boundary temperature of any kind of ice-rich media in the long term. The intensity of free convection in porous media depends strongly on the Rayleigh number which in turns depends on the permeability and soil layer dimensions. So, further investigation is needed to address this effect. Of course, according to your cited research, free convection is negligible for some kind of soils close to surface with specific permeability but it may still be important for other soil types with different thermal conditions like the study published more recently by Najafian Jazi et al (Jazi et al., 2024). We tried to better express this in the manuscript.

*"The intensity of free convection in soil depends strongly on the Rayleigh number which in turns depends on the permeability and dimensions of soil layer making it negligible, e.g. (Kane et al., 2001) or significant, e.g. (Jazi et al., 2024) with respect to the total heat transfer."*

**L335-338: Free convection is everywhere so it is beautiful, ok but not super relevant for your study. Funny that you did not explain the TC reader what is Boussinesq approximation but you do explain what is an iceberg 😌.**

The explanations about Boussinesq approximations were added.

**In this discussion, since your main conclusion is that we need to be careful about not underestimating the melt rates in rock massifs with ice cleft, I missed a bit of large scale discussion on the implication for:**

- **catchments water balance. If you try to upscale your results, how can this impact runoff in mountain catchments, river flow, lake levels, at catchment scale and a global scale? Where should we start worrying more about this question?**

- **Permafrost disappearance at the scale of the massif. Does it change what we forecast for the Alps, by much?**

**So that we can grasp how significant these results could be at broader scales.**

Thank you for your suggestions. This study is only a first step to underline the significance of convection on melting rates. Analyzing its impact at a regional/global scale is out of scope of this paper. Underlining that permafrost degradation along rock clefts could be enhanced by an order of magnitude as compared to classical models based on conduction, however, paves the way for investigating specific case studies.

**L345: "only impact on the temperature" I suspect the "on" should be removed.**

corrected:

**Reference**

Bartolomé, M., Cazenave, G., Luetscher, M., Spötl, C., Gázquez, F., Belmonte, Á., Turchyn, A. V., López-Moreno, J. I., and Moreno, A.: Mountain permafrost in the Central Pyrenees: insights from the Devaux ice cave, Cryosph., 17, 477–497, 2022.

Bergman, T. L., Lavine, A. S., Incropera, F. P., and DeWitt, D. P.: Fundamentals of Heat and Mass Transfer, Wiley, 2017.

Duvillard, P.-A., Ravanel, L., Schoeneich, P., Deline, P., Marcer, M., and Magnin, F.: Qualitative risk assessment and strategies for infrastructure on permafrost in the French Alps, Cold Reg. Sci. Technol., 189, 103311, 2021.

Fohlmeister, J., Luetscher, M., Spötl, C., Schröder-Ritzrau, A., Schröder, B., Frank, N., Eichstädter, R., Trüssel, M., Skiba, V., and Boers, N.: The role of Northern Hemisphere summer insolation for millennial-scale climate variability during the penultimate glacial, Commun. Earth Environ., 4, 245, 2023.

Ford, D. and Williams, P. D.: Karst hydrogeology and geomorphology, John Wiley & Sons, 2007.

Guyon, E., Hulin, J. P., Petit, L., and Mitescu, C. D.: Physical hydrodynamics, Oxford university press, 2015.

Jazi, F. N., Ghasemi-Fare, O., and Rockaway, T. D.: Natural convection effect on heat transfer in saturated soils under the influence of confined and unconfined subsurface flow, Appl. Therm. Eng., 237, 121805, 2024.

Jin, X.-Y., Jin, H.-J., Iwahana, G., Marchenko, S. S., Luo, D.-L., Li, X.-Y., and Liang, S.-H.: Impacts of

climate-induced permafrost degradation on vegetation: A review, Adv. Clim. Chang. Res., 12, 29–47, 2021.

Kane, D. L., Hinkel, K. M., Goering, D. J., Hinzman, L. D., and Outcalt, S. I.: Non-conductive heat transfer associated with frozen soils, Glob. Planet. Change, 29, 275–292, 2001.

Luetscher, M., Boch, R., Sodemann, H., Spötl, C., Cheng, H., Edwards, R. L., Frisia, S., Hof, F., and Müller, W.: North Atlantic storm track changes during the Last Glacial Maximum recorded by Alpine speleothems, Nat. Commun., 6, 6344, 2015.

Rohsenow, W. M., Hartnett, J. P., and Cho, Y. I.: Handbook of heat transfer, Mcgraw-hill New York, 1998.

Tubini, N., Gruber, S., and Rigon, R.: A method for solving heat transfer with phase change in ice or soil that allows for large time steps while guaranteeing energy conservation, Cryosph., 15, 2541–2568, 2021.

Virag, Z., Živić, M., and Galović, A.: Influence of natural convection on the melting of ice block surrounded by water on all sides, Int. J. Heat Mass Transf., 49, 4106–4115, 2006.

Walvoord, M. A. and Kurylyk, B. L.: Hydrologic impacts of thawing permafrost—A review, Vadose Zo. J., 15, 2016.

---

## Author Comment (AC2)

**GENERAL COMMENTS**

**The manuscript presents numerical analysis of the thawing of ice-saturated cleft in rocks, demonstrating the relative importance of gravity-driven convection of liquid water due to the subtle increase in density as the temperature rises from 0 to 4 degrees. While it is not difficult to imagine the significance of free convection based on observations in other environments, it is important to advance the quantitative understanding of the effects of free convection in ice-filled clefts. Therefore, this study has the potential to make a significant new contribution to cryospheric sciences. The manuscript is reasonably well organized and written in a clear language. However, it is missing some essential information on the methodology and as such, it is difficult for the reviewer to evaluate the rigor of numerical and experimental methods in some places. Theoretical interpretation of numerical modelling results are sound except for some missing details (see above), but the comparison between numerical results and field observation is much weaker. To strengthen the comparison, I suggest that the authors consider the following: (1) enhance the description of field methods, (2) acknowledge the magnitude of uncertainty in flow measurements more specifically, and (3) use independent evidence to support the match between model results and field observation. I will elaborate more on these in my specific comments below.**

We thank the reviewer for his constructive comments which will improve the quality of the manuscript. Since many questions of the reviewer deals with the model definition, we copy and paste the new version of the governing equation section that we tried to clarify. Our detailed answers to the specific questions of the reviewer follow. A list of reference cited in our answers is provided at the end of this document.

####################################################################################

NEW SECTION 2.2

*"2.2 Governing equations*

[revised manuscript text omitted]

*The boundary conditions are as follows. At the interface between an impermeable solid and a viscous fluid, the fluid velocity is equal to that of the solid (Guyon et al., 2015). This is the so-called no-slip and impermeability conditions, resulting in $u = v = 0$ at the rock-water interface. The temperature continuity and the heat flux conservation through this interface are also considered (since the water velocity vanishes at the rock-water interface, the heat flux through the interface reduces to conduction). As already mentioned in section 2.1, the bottom and vertical external boundaries are adiabatic, and the temperature evolution of the top boundary is imposed (see Fig.2b). "*

######################################################################

ANSWERS TO SPECIFIC QUESTIONS

**Line 18-19. The agreement the model and real-world observations is qualitative at best (see my comment on Line 273). 'The close agreement' is an overstatement. Please rephrase the sentence.**

corrected:

*« The model outcomes are compared qualitatively with field data from Monlesi ice cave (Switzerland) and confirm the agreement between real-world observations and the proposed model when free convection is considered. »*

**Line 91. Eq. 1 assumes no volume change in water, implying that the change in volume associated with ice-to-liquid transition is neglected in the analysis. If this is the case, then the model domain will have void space, presumably at the top. How does the model take this into account? Please present a clear explanation.**

It is true that the contraction due to ice melting is neglected in our model. This is equivalent to assume that an external water flow replenishes the top layer domain with water at the same temperature as the atmosphere. This would result in the additional vertical velocity in the liquid phase (Heitz and Westwater, 1970):

$$v_l = \frac{(\rho_l - \rho_s)}{\rho_l} \frac{dH}{dt}$$

This velocity would be that of the liquid in the absence of free convection, or would be added to the free convection velocity field in the other case. This additional flow can be safely neglected if the heat advected in that way is negligible compared to the heat absorbed by the motion of the melting front:

$$\rho_l \, v_l \, c_{pl} (T_{atm} - T_0) \ll L_m \rho_s \frac{dH}{dt}$$

Both equations above yield the condition of validity:

$$\frac{(\rho_l - \rho_s)}{\rho_s} \frac{c_{pl}(T_{atm} - T_0)}{L_m} \ll 1$$

With $(\rho_l - \rho_s)/\rho_s \simeq 0.09$, $c_{pl} \simeq 4200$ J/(kg.K), $T_{atm} - T_0 =$15°C and $L_m \simeq 334$ kJ/kg, the LHS of the above equation is approximately equal to 0.02, much lower than 1. The flow due to contraction can thus be safely neglected. This analysis will be inserted in the new version of the paper.

Heitz and Westwater (1970) display an interesting comparison of mathematical solutions with equal and unequal phase densities for the freezing of water initially at 13.8°C, with top boundary suddenly decreased at -46.5°C. The melting front velocities computed from both models are in very good agreement.

**Line 92. Reduced pressure. This term is not familiar to most readers of the journal. Please define it**.

The momentum equations have been rewritten to avoid the use of the reduced pressure. In the new version of the manuscript, $p$ is the fluid pressure (this modification is purely formal, the model is unchanged).

**Line 94. A kind of Darcy-like pressure drop. I do not understand what this means. Please explain it more clearly.**

The Darcy law is $u = -\dfrac{k}{\mu}\nabla p$ , k is permeability and $\mu$ is dynamic viscosity. If we rewrite this equation like $\nabla p = -\dfrac{\mu}{k}u$ , there is a similarity between this pressure gradient and the term $A\dfrac{(1-\theta)^2}{\theta^3 + \varepsilon}u$ in the momentum equations. In fact, the term $A\dfrac{(1-\theta)^2}{\theta^3 + \varepsilon}$ and $\dfrac{\mu}{k}$ has the same unit and that's why it is called often Darcy-like pressure drop (or gradient) in the literature (specially in numerical modeling of melting and solidification)(Nazzi Ehms et al., 2019). However, this is a technical (and quite complicated) aspect of the numerical method not necessary to understand the paper. In the new version, the presentation has been simplified. We now focus on the main principles of the numerical method, and we refer the reader to the literature for more details (see the new version of section 2.2 above).

**Line 96. What does the variable 'A' physically represent? What is the unit?**

The variable *A* is a numerical parameter with no physical sense. In the new version, we tried to clarify the distinction between physical and numerical parameters (see the new version of section *"2.2 Governing equations" above*).

**Line 109. The heat transfer in the rock phase is limited to conduction, implying negligible rock porosity. This is contrary to numerous field-based studies of karst hydrogeology, where the fracture network in karstified rocks can play a major role in water transfer. What is the expected porosity of the system the authors are intending to model? Please add a sentence or two on porosity and fractures.**

Thank you for your feedback. We assumed an impermeable (solid) rock massif containing one ice-clefts with an aperture size ranging between 2 to 50 cm (10 cm as a reference value in figure-1). These figures are consistent with field observations and correspond to a karst porosity of between 0.2 and 5 %, depending on the density of fractures. Our study is not relevant for smaller fractures with lower aperture sizes, where the effect of free convection is expected to be less significant. The thermal impact of water flow on the surrounding rock temperature will be addressed elsewhere.

**Line 122. The authors assume -0.5 to +0.5 C as the temperature rage of ice-liquid transition. While ice and liquid water can co-exist under negative temperature due to freezing-point depression, there is no known mechanism to sustain the ice-liquid mixture under positive temperature. Please justify the choice of temperature range. If it is not justifiable, please re-run the simulations using a physically feasible temperature range.**

This melting temperature range is a numerical parameter, with no physical sense. See the second paragraph of the new version of section "*2.2 Governing equations*" above. We added a paragraph to explain how its numerical value was chosen:

*"Regarding the numerical parameters required to model melting, we imposed ΔT=0.5 ℃, A =1000 kg.m$^{-3}$.s$^{-1}$ and $\varepsilon = 10^{-3}$. We checked that imposing $\Delta T = 0.3$ ℃ or 0.7 ℃ did not significantly change the results (see Appendix A). The selected values of $A$ and $\varepsilon$ produced vanishingly small velocity fields in the ice with no deterioration of the solver stability."*

**Line 127. Impermeable solid rock. Please see my comment on Line 109.**

See our answer to the question about line 109.

**Line 132. How is the depth of the cleft defined with respect to the actual clefts in the field. In natural systems, water will drain from the bottom of the cleft as shown in Figure 1a.**

The following paragraph has been added in the introduction:

*"We consider the upper part of a single vertical cleft of size aperture $A_p$ filled with pure water whose melting point is $T_m = 0°C$. This cleft is surrounded by a rock mass of width $W$ (see Fig.1b). In karst massifs, water flow concentrates in well-defined conduits (Ford and Williams, 2007). The micro-porosity of the rock is thus disregarded, and impermeable rock mass is assumed.*

*The cleft is located at the center of the 2D domain of height $H_{dom}$. In the initial state, the system (water and surrounding rock) is at the uniform initial temperature $T_i$=-1°C, and all the water is frozen. At time t=0, the temperature of the ground surface $T_s$ increases at the constant rate 1.77 ℃/hour to reach 15°C after 9 hours. This temperature increase is similar to the daily warming between the early morning and the afternoon.*

*The effect of the cleft aperture size was investigated by varying $A_p$ from 2 cm to 50 cm. We imposed $H_{dom}$=0.8 m and W =1 m in all simulations. These values are large enough so that the thermal perturbation induced by the presence of the cleft does not reach the domain boundaries at the end of the simulated time (9 hours). The vertical and bottom boundaries of the domain can therefore be considered as adiabatic (see Fig.1b). It is important to note that the domain height $H_{dom}$ contains only the upper part of the cleft, whose actual depth commonly ranges from 1 to 10 m. The value of $H_{dom}$ used in this study is convenient for the daily time scale considered in the numerical simulations. Simulating larger time scales would require larger values of $H_{dom}$.*

[Figure]

Fig. 1-b

**Line 132. 80 and 10 cm. How are these values chosen? Please provide an explanation.**

Please see above our previous reply for Line 132.

**Table 1. Density 2320 kg/m3. This is much smaller than a typical density of solid carbonate rocks, and implies substantial porosity (14%?). Is this consistent with the model assumption? Please explain.**

**Table 1. Thermal conductivity 1.656 W/m/K. This is much smaller than that of solid carbonate rocks, implying substantial porosity. Is this consistent with the model assumption?**

The assumed rock density and heat capacity were taken from (Covington et al., 2011). The thermal conductivity is from (Guerrier et al., 2019). The table was updated.

The literature reports a broad range of thermal properties. Wenk and Wenk, (1969) reported a density range between 2510 and 2840 [kg/m$^3$] and a thermal conductivity range between 0.97 and 1.99 [W/m/K] for carbonate alpine rocks and Zappone and Kissling, (2021) obtained a density range between 2150 and 2823 [kg/m$^3$] for limestone. To test the model sensitivity to rock properties, we run a simulation with a density value of 2700 [kg/m$^3$] and thermal conductivity of 2.2 [W/m/K] (Luetscher et al., 2008). The fig. R2-1 displays the melting rate as a function of time for modified (new values of rock density and thermal conductivity) and unmodified (initial) properties. These results show that after 9 hr the melting rates only differs by 10%.

[Figure]

Fig. R2-1 Melting rate for modified and unmodified rock thermal properties

**Table 1. Thermal properties and numerical parameters. The liquid water properties are temperature dependent. The properties of ice and rock are assumed constant.**

| Thermal properties | values | Reference |
|---|---|---|
| $\rho_s$ (kg/m³) | 916.2 | - |
| $k_s$ (W/m/K) | 2.22 | - |
| $c_{p,s}$ (J/kg/K) | 2050 | - |
| $\rho_l$ (kg/m³) | see Fig.2 | (Comsol, 2018) |
| $k_l$ (W/m/K) | 0.556 at 0°C 0.585 at 15°C | (Comsol, 2018) |
| $c_{p,l}$ (J/kg/K) | 4216 at 0°C 4192 at 15°C | (Comsol, 2018) |
| $\mu$ (mPa.s) | 1.79 at 0°C 1.43 at 7.5°C 1.15 at 15°C | (Comsol, 2018) |
| $\rho_r$ (kg/m³) | 2320 | (Covington et al., 2011) |
| $k_r$ (W/m/K) | 1.656 | (Guerrier et al., 2019) |
| $c_{p,r}$ (J/kg/K) | 810 | (Covington et al., 2011) |
| $L_m$ (J/kg) | 334000 | - |

[Figure]

Figure 2: density of liquid water as a function of temperature.

**Line 141. Please spell out 14k, 24k, etc.**

It was corrected: *"The total number of elements in each of the four cases was almost 14000, 24000, 32000, and 47000, respectively"*.

**Line 179. At the bottom of the cavity. The model also under-simulates the advance of thawing front in the upper part by 2300 sec. Please point this out.**

It is difficult to point out what specific assumptions cause this discrepancy. The following sentence has been added in the manuscript:

*"Although some discrepancies exist between the experiments and the numerical model, especially at the bottom of the cavity at the start of the simulation and in the upper part at later times, the overall performance of our model is suitable to represent the free convection cells and their effect on ice melting despite the simplifying assumptions made in the model (including 2D geometry and negligible volume contraction upon melting)."*

**Line 170. The overall performance of our model is sufficient. This is a subjective statement. Please explain the basis for this statement.**

The sentence was modified (given in previous comment Line 179).

**Line 188-189. The authors state that the model conceptualization (Figure 1b) is similar to the physical setting depicted in Figurer 1a. However, I do not see a clear similarity. Please improve the description. A schematic diagram depicting typical clefts observed in the field will be useful to bridge the gap between Figures 1a and 1b.**

This comment is related to Line 132. Figure-1-b was updated and better descriptions about the differences of computational domain and the real ice-clefts were added to the manuscript.

**Line 193. Total volume of liquid water. The total volume of liquid water should be much smaller than the volume of ice before melting. Therefore, there should be some void spaces if the model obeys the mass-conservation law. Violation of mass conservation is considered a major deficiency of any mass and water transfer models. Please explain how the water mass is conserved in the model.**

See our answer to the question related to line 91.

**Line 223. Rayleigh number. Please report the Rayleigh numbers computed for the numerical experiments presented in the manuscript.**

The depth of meltwater after 9 hr is about 0.3 m and the resulting Rayleigh number is in the order of $10^8$. This information has been included in the new version.

**Line 230. Melting rate. Does it refer to the melting rate (kg/hr/m2) or cumulate amount of melt (kg/m2)? Linear plots in Figure 10 seem to suggest the latter. Please clarify.**

The legend of Fig.10 is correct. The melting rate actually increases with time. This clarification has been included in the text.

**Line 244. 8 m meltwater depth. Is this 8 m or 0.8 m? The model has 0.8 m, not 8 m. Please clarify.**

See our answer to the question related to line 132.

**Line 253. Seasonal freezing seals them periodically. This implies that the clefts (or chimneys) are not always saturated. How can this be adequately represented in the saturated model? Please explain. By the way, are clefts and chimneys the same thing? If so, please use a consistent term.**

Chimneys and clefts refer to different geometries. For the purpose of this paper, we can indeed focus the description only on clefts. The sentence was corrected accordingly.

The monitored site freezes seasonally from the bottom, rising the hydraulic head within the cleft. With further freezing the water within the cleft will completely solidify until the next thawing season. We believe our model is the best possible analogue for approaching the complexity of this natural system.

*"The c. 600 m2 cave chamber is partly filled with perennial congelation ice fed by a number of vertical clefts of between $10^1$ and $10^3$ mm width"*

**Line 254. The distance. Does this refer to vertical distance? Does the external surface indicate the ground surface? Please clarify. A schematic diagram will be useful (see my comment on Line 188-189).**

Yes. It was modified as follows: *"The vertical distance between the external ground surface and the cave ceiling reaches c. 20 m …"*

**Line 255. Clefts of different sizes. Please indicate a rage of sizes observed at the site.**

The sentence was modified as follows: *"[…] fed by a number of vertical clefts of between $10^1$ and $10^3$ mm width."*

**Line 256. A few centimeters. Please report the actual depth, even if it is approximate.**

Modified : *" […] soil temperature at 10 cm depth "*

**Line 257. 4.5 days. Please report the actual dates.**

The date of measurement was added in new version of manuscript: *"from April 13 to 17"*

**Line 259. Cave temperature. Where in the cave (in relation to clefts) and how was it measured? Please explain.**

**Line 260. How was the water flow monitored? This information is critical. Please explain it carefully with sufficient details.**

We added two paragraphs for measuring cave temperature and water flow rate:

*"The main water inlet at Monlesi (subcutaneous flow) was instrumented to measure discharge rates at 30 min intervals using a pressure probe set at the bottom of a 1 m long perforated PVC tube capturing the inlet. The water height measured in the tube is converted to discharge (Q, in l/min) with an empirically calibrated rating curve checked by nine manual "bucket gauging" between 0 and 13 l/min with an uncertainty of ±10% (Luetscher et al., 2008).*

*Cave air temperatures were measured using negative temperature coefficient thermistors with a resistance of c.29.5 kOhm at 0°C and a temperature coefficient of about 5% °C-1 (YSI 44006). The thermistors were calibrated in a bath of melting ice to an accuracy of ±0.1°C. The thermistors were spaced at 2 m intervals in two chains comprising 19 sensors dispatched in the main cave chamber (Luetscher et al., 2008). Air temperatures were recorded at 1 h intervals and logged externally on a Campbell CR10X data logger with two multiplexer logging units."*

**Line 264-265. I do not exactly understand this sentence. 'We allow for the meltwater to drain deeper' implies that a new model is set up to simulate drainage, but 'in contrast to our model' implies that the same model without drainage is still used. Please clarify. Also, if drainage is allowed, please explain how it is done in the model.**

This sentence has been deleted in the course of the revision.

**Line 271. 3 meters depth. Is this based on the measured depth in the field? If not, how is it selected? The same applies to 10 cm aperture.**

The details about the selection of the total depth of the domain were explained above (Line 132). The total depth of meltwater considered in the model depends on the duration of the warming which is about 4.5 days here (in section 4.3). This figure compares to the previous cases which considered a duration of 9 hours in the two predefined scenarios shown in section 4.1 and 4.2. The depth of the entire domain should be sufficiently high to ensure that its basis remains continuously obstructed by ice. In the field, this is confirmed by the negative temperature measured in the cave. In other words, the meltwater should exit the domain from somewhere above the ice interface although this outlet is not considered in our conceptual model. We changed a little the corresponding sentence for more transparency:

*"A rough estimate of the cavity geometry (computational domain) assumes an initial ice-filled cleft with 3 meters depth ($H_{dom}$=3 m) and 10 cm aperture size ($A_p$=10 cm) subject to a linear temperature rise at the top boundary (black dash-dotted in Fig. 11-b)."*

**Line 271. In general, how long are the clefts observed at this site? Please indicate it in the sentence. I am referring to the third dimension in addition to the depth and the aperture.**

The typical length of these clefts reaches an order of 1 m. The sentence was amended accordingly.

**Line 273. The same order of magnitude. This is not a meaningful comparison because the measured flow rate may have a large degree of uncertainty depending on how it was measured (see my comment on Line 260). It is not uncommon for this kind of measurements to have uncertainties greater than an order of magnitude. This is a major weakness of the manuscript. Independent evidence demonstrating the qualitative match between the model and field observation will be useful (see my comment on Line 311-312).**

The uncertainty on the measured flow rate is ±10 % (Luetscher et al., 2008). This figure is now mentioned in the revised manuscript (cf also response to comment on Line 260)

**Line 290. Sufficiently high. Please indicate the number and compare it with the critical Rayleigh numbers reported in previous studies of free convection (not necessarily in water-ice systems).**

This section has been rewritten as follows:

*"In the present work, we simulated 9 hours of atmosphere temperature increase. When the aperture size $A_p$ was varied from 2 to 50 cm, the liquid height H at the end of the simulation approximately ranged from 30 to 40 cm, and the convection cell occupied the entire liquid domain. However, the liquid height reached after 9 hours is only a small part of the actual height of the cleft (commonly up to 10 m). H is expected to increase if longer times are considered. The question arises whether the free convection cells always fill the entire liquid domain at longer times, despite the increase of friction due to lower aspect ratio $A_p$/H.  If the convection cell occupies only a part of the cavity, the efficiency of heat transfer between the ground surface and the melting front will be reduced.  The significance of free convection can be assessed from the value of the dimensionless Rayleigh number*

$$Ra = \frac{g\beta(T_c - T_H)H^3}{\alpha_l \, \nu_l} \qquad (10)$$

*where ($T_c$-$T_H$) is the temperature difference between bottom and top surfaces, $\alpha_l$ and $\nu_l$ are the liquid water diffusivity and kinematic viscosity, respectively. Ra represents the ratio of the diffusion time over the free convection time ($Ra\sim 10^8$ in the numerical experiments presented in this article). In a cavity with infinite lateral dimensions, free convection is triggered when $Ra \gtrsim 10^3$ (otherwise, the conductive state is stable, see Bergman et al (2017) for more information about the Rayleigh-Bénard instability). However, in the confined geometry considered in this work, the presence of the vertical walls must be considered. Rohsenow et al (1998) provide the following condition for convection onset, which takes into account the stabilizing effect of the vertical walls for $A_p \ll H$, in the limiting case of perfectly conducting walls:*

$$Ra \gtrsim 10^2 \times \left(\frac{H}{A_p}\right)^4 \qquad (11)$$

*Injecting Eq.(10) in Eq.(11) yields the maximum value of the liquid height H for which the free convection cell extends from the ground surface to the melting front:*

$$H \lesssim 10^{-2} \times \frac{g\beta(T_c - T_H){A_p}^4}{\alpha_l \, \nu_l} \qquad (12)$$

*Considering that the liquid region at temperature T>4°C is stable and that the isotherm 4°C is close to the top of the cleft when the free convection cell fills the entire cavity (see Fig. 7b), we get ($T_c$-$T_H$)=-4°C. Using the physical properties from section 2.3 and $A_p = 2$ cm (the minimum aperture size considered in this study) yields $H \lesssim 10$ m, which is also the order of magnitude of the maximum cleft height. Therefore, free convection cells should always extend throughout the melted region for $A_p \gtrsim 2$ cm. Note that the assumption of perfectly conducting walls used in Eq.(11) is less favorable to convection than finite conductivity (Rohsenow et al, 1998). Eq.(12) is thus expected to slightly underestimate the higher bound of H corresponding to fully developed free convection cells."*

**Line 311-312. The modeled melting rate alone is not sufficient to support the model performance due to the large degree of uncertainty in flow measurements. Independent evidence will be useful. For example, how long does it usually take to thaw a cleft completely from the top to the bottom? Will it be possible to estimate the thawing rate and compare it with the modelled thawing rates? Please explore this and other approaches further to provide independent evidence.**

The uncertainty on the flow measurements is low compared to the uncertainties on the cleft geometry and we do not consider this as an issue. Providing a more robust estimate on the thawing rate is impossible in absence of an exact estimate of the ice volume hosted in the cleft. Nonetheless, one can consider that the cleft is completely thawed if the water flow responds to rainfall after a period of drought. This case is observed on April 30, 2003, 13 days after the end of our monitoring period. Assuming the cleft (3x0.1x1 m) was completely saturated with ice this yields a maximum thawing rate of 0.3m3/350h = 0.9 l/h to be compared with a measured flow rate averaging 0.5 l/h.

**Line 447. At 8 m meltwater depth. Where does 8 m come from? The numerical model was 0.8 m deep, not 8 m.**

See our answer to the question related to line 132.

---

## Author Response (AR2)

We thank the referee for his careful reading of the manuscript, which has been modified according to his remarks. In the following, the referee comments are in bold characters, the modifications made in the manuscript are in italic.

**I am generally happy with the changes made by the authors. I think the paper is more accessible, the approach is clearer and explained in a more pedagogical way. I have only very minor suggestions that I would like to take into account, but otherwise I think the manuscript is generally ready for publication.**

**L132 "effective properties"**

**Does this mean model variable? Is it a common denomination?**

"Effective properties" refer to the physical properties to be considered in the simulation. The term "effective" means that they may depend on some physical assumptions made in the governing equations (e.g., the Boussinesq approximation or the specific model used for the change of state). This is a common terminology. We modified the sentence:

*"No subscript indicates the effective physical properties to be considered in the governing equations of the water domain (solid, liquid and diphasic)."*

**L174: Maybe repeat the reference of Comsol (2018) in the figure caption.**

Done.

**L193: Here again I would insist, "[they]investigated with an experimental approach" or something similar. To remind the reader that it's a comparison with experimental observations.**

Corrected:

*"Virag et al., (2006) investigated with an experimental approach the effect of free convection on …"*

**L209: "4.1 Stagnant liquid water (SLW) versus free convection (FC)" not formatted as a title**

Thank you for pointing this out. We corrected it in the manuscript.

**L218: "an extreme slope" the wording is a bit peculiar, it suggests something crazy is happening. Maybe reword.**

The corresponding paragraph has been reworded as follows.

*"When convection is disregarded (SLW), the melting front is nearly horizontal except close to the walls, where a steep slope is observed. The higher thermal diffusivity of the rock ($\alpha_r \approx 8.8 \times 10^{-7}$ $m^2/s$) compared to that of the liquid water ($\alpha_l \approx 1.3 \times 10^{-7}$ $m^2/s$) results in faster heat propagation in the rock, and enhanced melting of the ice closer to the rock. When free convection is considered (FC), the advection of heat by the flow results in faster propagation of the melting front, with an inversion of its curvature (the meting front propagates faster in the center of the cleft than close to the walls)."*

**L231-232 "This shows that the circulation of water inside the cleft results in a thermal bridge between ice interface and top atmosphere."**

**Isn't it exactly the point of including free convection? It's in the model in order to do that no?**

We modified the sentence: *"This confirms that the circulation…"*

**Figure 9: It is a relatively common practice to "turn" graphics which have depth as an axis so that depth is vertical.**

We followed the reviewer's suggestion and modified the figure accordingly.

**L258: "heat flux propagates perpendicularly to the cleft"**

**So horizontally? I don't visualize.**

We tried to clarify this point as follows.

*"For the SLW case, heat transfer is mainly driven by diffusion in the rock, which has a greater diffusivity than water (see the temperature contour in Fig.8a). At a given depth, the rock is warmer than the water. The ice directly in contact with the rock thus melts faster. This explains the larger specific melting rate obtained with the smallest aperture ($A_p$=2 cm)"*

**L319: I would add "similar to the soil temperature increase" when describing the T forcing**

We modified the sentence as follows:

*"To assess the melting rate, we considered the same conceptual model as in Fig.1. A linear temperature rise from 0 to 5°C during 4.5 days was assumed at the top boundary (black dashed-dotted line in Fig.12b). This temperature rise is similar to the temperature evolution measured in the soil (solid black line in Fig.12b)."*

**L327: "The cleft is subject to a linear temperature rise" redundant to line 318-319 if I'm correct.**

We deleted this redundant sentence.

**L333-344 do not discuss anything, they wrap up the study like a conclusion I feel.**

This paragraph has been moved in the conclusion, which has been reorganized as follows:

*"A quantification of the melting rate of ice-rich permafrost in heterogeneous media is essential to assess the speed of permafrost degradation. Our model relies on a 2D approach coupling free convection (buoyancy-driven flow) in a vertical cleft with conduction in the surrounded homogeneous rock. Increasing the temperature of the ground surface can generate free convection cells because water-density increases between 0 and 4°C (a property specific to water). The convection cells generate a thermal bridge between the atmosphere and the melting front, resulting in the formation of a mixing*

*zone with quasi-uniform temperature in the water column. This dramatically enhances the melting rate of interstitial ice when compared to models assuming stagnant liquid water (about an order of magnitude after 9 hours for an aperture size of 10 cm). In contrast to scenarios assuming conduction in stagnant liquid water, for which the temperature signal from the atmosphere is fully attenuated beyond a certain distance known as the diffusion length, the presence of free convection extends over greater distances. This thermal penetration also exerts an influence on the surrounding rock. Despite simplifying assumptions in the model and many uncertainties about the cleft geometry and the measured water flow rate, melting rate predicted by a model including free convection fit the order of magnitude measured in Monlesi cave (Fig. 12).*

*The significance of free convection should also be estimated in similar thermal configurations with different geometries such as cylindrical conduits or 3D cavities. Furthermore, the effect of free convection is not limited to hourly or daily oscillations and can be studied over much longer timescales, including centennial to millennial fluctuations. Currently, the computational costs are the main barriers for including free convection in long-term simulations. The full coupling of the momentum and energy equations requires the time steps being much smaller compared to simple conduction-based models. Further investigations are thus ongoing to reformulate the governing equations and simplifying them for simulations over longer time-scales. Moreover, refreezing processes have yet to be considered to fully represent the long-term evolution of such a system.*

*Eventually, the effect of water free convection on ice melting rate is not limited to permafrost regions. For instance, the melting of icebergs can also be impacted by water free convection (Couston et al., 2021; Hester et al., 2021) increasing production of freshwater in oceans with potential impacts on the climate at global scale."*

**L374: "Forced convection"**

**Would it be forced convection if the water comes from outside the domain? I would have said that forced convection happens when you stir in one given place, not when you bring water from somewhere else, I would call it advection. I am happy to be proven wrong though.**

We feel that forced convection is correct in this context. In the field of heat transfer, "forced convection" (the flow is due to an external force) is opposed to free convection (the flow is due to buoyancy). The term "forced convection" applies equally to opened systems (as heat exchangers) or closed systems (as stirred-tank reactors). See for instance Bergman et al., (2017).

**L403: Figure A1. Effect on the melting temperature range $\Delta T$ of the melting?**

Corrected.

**Reference**

Bergman, T. L., Lavine, A. S., Incropera, F. P., and DeWitt, D. P.: Fundamentals of Heat and Mass Transfer, Wiley, 2017.

Couston, L.-A., Hester, E., Favier, B., Taylor, J. R., Holland, P. R., and Jenkins, A.: Topography generation by melting and freezing in a turbulent shear flow, J. Fluid Mech., 911, A44, 2021.

Hester, E. W., McConnochie, C. D., Cenedese, C., Couston, L.-A., and Vasil, G.: Aspect ratio affects iceberg melting, Phys. Rev. Fluids, 6, 23802, 2021.

We thank the referee for pointing out the passages of the manuscript which require clarification. We hope that the clarity of the revised version has been improved. In the following, the referee comments are in bold characters, the modifications made in the manuscript are in italic.

**The authors have sufficiently addressed many of my comments on the original manuscript. However, there are still a few issues that have not been addressed in a satisfactory manner. I feel that these issues need to be addressed before the manuscript is considered for publication in the journal. Please see my specific comments for the remaining issues. The line numbers indicate those in the original manuscript.**

**SPECIFIC COMMENTS**

**Line 91. This was the comment concerning the volume change during ice-to-liquid transition. Eq. 2 in the revised manuscript still assumes no volume change in water, implying that the change in volume associated with ice-to-liquid transition is neglected in the analysis. The authors presented an explanation for how this may be interpreted in their response, but it is not explained in the text. Omission of volume change is a major assumption in the system equation, and it needs to be explicitly justified in the text. Please add a convincing explanation to the texts.**

Regarding volume changes, there are two distinct assumptions (independent from each other):

1) We neglect the change of volume induced by ice melting (i.e., the difference of density between ice and liquid water). We show in section 2.3 that the validity of this assumption depends on condition (14), and that this condition is satisfied for the problem that we investigate. We also provide a reference (Heitz and Westwater, 1970) showing that this assumption has no significant effect on the results, in a configuration similar to ours. We believe that the validity of this assumption has been fully justified in the text.

2) For the liquid phase, we apply the standard Boussinesq approximation, which consists in assuming constant liquid density in all the terms of the governing equations (including the mass conservation equation (2)), except in the buoyancy term of the momentum balance. Although this approximation may appear to lack rigor, it is very classical and widely used for modeling of free convection. It has been first stated by the French physicist Joseph Boussinesq at the end of the 19$^{th}$ century, and its self-consistency has been mathematically demonstrated later (see for instance the reference Spiegel and Veronis (1960) added in the revised version).

In order to clarify these points, we reorganized the passage of section 2.3 that deals with volume variations:

*"The standard Boussinesq approximation is assumed in the liquid phase (Spiegel and Veronis, 1960) (Bejan, 2013). This approximation, widely used for free convection modeling, consists in assuming constant liquid density in the governing equations Eqs.(2-4), except in the buoyancy term of the momentum balance equations Eqs.(3-4). The thermal expansion coefficient $\beta$ at the origin of buoyancy is estimated from the relation*

$$\beta = -\frac{1}{\rho_l}\frac{d\rho_l}{dT} \qquad\qquad (11)$$

*where $\rho_l$ is the temperature-dependent liquid water density displayed in Fig.2 (the order of magnitude of $\beta$ in the unstable temperature range from 0 to 4°C is approximately $-3 \times 10^{-5}\ K^{-1}$). In contrast,*

*the constant liquid density $\rho_0$ estimated at the reference temperature $T_0$ is considered in the inertia terms of Eqs.(3-4) ($\rho_0 = 999.84$ kg/m³ at $T_0 = 0°C$). The Boussinesq approximation is valid if the maximum fluid density variation $\Delta\rho_l$ is much lower than the liquid density $\rho_l$, a condition usually satisfied in liquids ($\Delta\rho_l/\rho_l \sim 10^{-3}$ in our case).*

*The density of water is greater than that of ice by approximately 10%. This induces a reduction of volume upon melting which is neglected in our model. This is equivalent to assuming that an external water flow replenishes the top layer domain with water at the ground surface temperature $T_s$. This would result in the additional vertical velocity in the liquid phase (Heitz and Westwater, 1970):*

$$v_l = \frac{(\rho_l - \rho_s)}{\rho_l} \frac{dH}{dt} \qquad (12)$$

*This velocity would be that of the liquid in the absence of free convection, or would be added to the free convection velocity field in the other case. This contraction-induced flow can be neglected if the heat advected in that way is negligible compared to the heat absorbed by the motion of the melting front:*

$$\rho_l \, v_l \, c_{pl}(T_s - T_m) \ll L_m \rho_s \frac{dH}{dt} \qquad (13)$$

*Eqs.(12-13) yield the condition of validity:*

$$\frac{(\rho_l - \rho_s)}{\rho_s} \frac{c_{pl}(T_s - T_m)}{L_m} \ll 1 \qquad (14)$$

*with the physical properties of table 1 and $T_s - T_m =15°C$, the LHS of Eq.(14) is approximately equal to 0.02, much lower than 1. The volume change induced by melting can thus be safely neglected. Heitz and Westwater (1970) presented a comparison of mathematical solutions with equal and unequal phase densities. In a configuration close to ours, they show that considering equal densities for ice and liquid water resulted in a negligible loss of accuracy."*

**Line 122. This was the comment concerning the temperature range, in which ice and liquid water co-exists. The original manuscript had the range of -0.5 to +0.5 C, which did not make physical sense because there is no known mechanism of keeping ice above 0 C. The authors responded by stating that the variable is a numerical parameter with no physical sense. However, this statement is incorrect because fluid density is a function of the actual temperature (Fig. 2 in the response). Density of liquid water at -0.5 C is different from that at +0.5 C. Such as small difference in density is usually not a problem, but in this particular case, it may have a noticeable effect because the system is driven by a small density gradient. Please investigate this further and present a convincing justification in the texts.**

Strictly speaking, the phase change transition over the melting range $[T_m - \Delta T, T_m + \Delta T]$ is valid for binary systems. However, when $\Delta T \rightarrow 0$, this kind of model asymptotically converges to the case of a pure substance with a phase change temperature at $T_m$. This is why models based on a melting range are widely used for pure substances (Michałek and Kowalewski, 2003; Zeneli et al., 2019; Bourdillon et al., 2015; Arosemena, 2018). When pure substances are considered, the choice of $\Delta T$ results from a compromise. Decreasing $\Delta T$ increases the model accuracy (the model gets closer to the ideal case $\Delta T = 0$), but requires more computational resources. Practically, we know that $\Delta T$ is small enough (i.e., the model is accurate enough) by checking that varying $\Delta T$ does not significantly change the

results. This is the purpose of appendix A, where it is shown that the model is a good approximation of a pure substance if $\Delta T \leq 0.7°C$ (decreasing $\Delta T$ below 0.7°C does not significantly modifies the results). This test implicitly validates that the change of density in the range $[T_m - \Delta T, T_m + \Delta T]$ has no significant impact on the results in this case. We tried to clarify this point in the new version:

*"To avoid the difficult task consisting in tracking the moving boundary between ice and liquid water, we adopted a strategy that allows to define the same set of dependent variables and governing equations in the entire water domain. To this end, we do the approximation of smooth phase transition between solid and liquid phases. We assume that ice melting begins at temperature $T_{m1} = T_m - \Delta T$ and ends at $T_{m2} = T_m + \Delta T$ (water is in solid state for $T < T_{m1}$, in liquid state for $T < T_{m2}$, and both phases coexist for $T_{m1} \leq T \leq T_{m2}$). It is important to note that in this study, $\Delta T$ is a numerical parameter with no physical meaning. Ideally, the behavior of a pure substance melting at temperature $T_m$ is recovered for $\Delta T \to 0$. Decreasing $\Delta T$ thus improves the model accuracy, but requires more computational resources (see (Michałek and Kowalewski, 2003; Zeneli et al., 2019; Bourdillon et al., 2015; Arosemena, 2018) for more details). Practically, the setting of $\Delta T$ results from a sensitivity analysis. Its value must be decreased until it does not change the results. This is the purpose of appendix A, where it is shown that the model is a good approximation of a pure substance for $\Delta T \leq 0.7°C$."*

**Line 260. This was about the method of flow monitoring and its uncertainty. The authors responded by indicating the uncertainty in flow measurement itself. However, from what I can infer from the texts, flow was measured at the main water inlet, which collects the total flow from multiple clefts and chimneys. Therefore, the flow measurement may not represent the actual flow form the particular cleft, to which model results are compared against. Please add more specific explanation about the measurement (or estimate) of flow from the particular cleft and its uncertainty.**

The water flow was monitored at the main inlet which originates **from one single cleft**. The defined computational domain (i.e. H$_{dom}$=3 m, Ap=10 cm, L=1m) approaches the known cleft geometry to the best of our knowledge.

Whether the actual cleft is itself fed by a network of smaller secondary fractures hidden in the host rock is beyond our knowledge and there is no way to address this issue empirically. This point was already discussed in l. 327-331. But overall, our measurements represent indeed the actual flow drained by the visible cleft. Measurement uncertainties (±10%) were already added to the previous revision.

Assuming there is a conceptual misunderstanding, we edited l. 309 to make this point even more clear: *"The main cleft was instrumented to measure discharge …"*

**Reference**

Arosemena, A.: Numerical Model of MeltingProblems, 2018.

Bejan, A.: Convection heat transfer, John wiley & sons, 2013.

Bourdillon, A. C., Verdin, P. G., and Thompson, C. P.: Numerical simulations of water freezing processes in cavities and cylindrical enclosures, Appl. Therm. Eng., 75, 839–855, 2015.

Caggiano, A., Mankel, C., and Koenders, E.: Reviewing theoretical and numerical models for PCM-embedded cementitious composites, Buildings, 9, 3, 2018.

Heitz, W. L. and Westwater, J. W.: Extension of the numerical method for melting and freezing problems, Int. J. Heat Mass Transf., 13, 1371–1375, 1970.

Michałek, T. and Kowalewski, T. A.: Simulations of the water freezing process–numerical benchmarks, Task Q., 7, 389–408, 2003.

Nazzi Ehms, J. H., De Césaro Oliveski, R., Oliveira Rocha, L. A., Biserni, C., and Garai, M.: Fixed grid numerical models for solidification and melting of phase change materials (PCMs), Appl. Sci., 9, 4334, 2019.

Spiegel, E. A. and Veronis, G.: On the Boussinesq approximation for a compressible fluid., Astrophys. Journal, vol. 131, p. 442, 131, 442, 1960.

Zeneli, M., Malgarinos, I., Nikolopoulos, A., Nikolopoulos, N., Grammelis, P., Karellas, S., and Kakaras, E.: Numerical simulation of a silicon-based latent heat thermal energy storage system operating at ultra-high temperatures, Appl. Energy, 242, 837–853, 2019.